# Bridging Implicit and Explicit Geometric Transformations for Single-Image View Synthesis

## Abstract

Creating novel views from a single image has achieved tremendous strides with advanced autoregressive models. Although recent methods generate high-quality novel views, synthesizing with only one explicit or implicit 3D geometry has a trade-off between two objectives that we call the "seesaw" problem: 1) preserving reprojected contents and 2) completing realistic out-of-view regions. Also, autoregressive models require a considerable computational cost. In this paper, we propose a single-image view synthesis framework for mitigating the seesaw problem. The proposed model is an efficient non-autoregressive model with implicit and explicit renderers. Motivated by characteristics that explicit methods well preserve reprojected pixels and implicit methods complete realistic out-of-view region, we introduce a loss function to complement two renderers. Our loss function promotes that explicit features improve the reprojected area of implicit features and implicit features improve the out-of-view area of explicit features. With the proposed architecture and loss function, we can alleviate the seesaw problem, outperforming autoregressive-based state-of-the-art methods and generating an image $\approx$100 times faster. We validate the efficiency and effectiveness of our method with experiments on RealEstate10K and ACID datasets.

## 1   Introduction

Single-image view synthesis is the task of generating novel view images from a given single image [5, 18, 23, 38–40, 47, 50, 54]. It can enable the movement of the camera from a photograph and bring an image to 3D, which are significant for various computer vision applications such as image editing and animating. To perform the realistic single-image view synthesis in these applications, we can expect that the novel view image has to consist of existing objects and unseen new objects from the reference viewpoint. Therefore, for high-quality novel views, the following two goals should be considered: 1) preserving 3D transformed seen contents of a single reference image and 2) generating semantically compatible pixels for filling the unseen region. To achieve two goals, explicit and implicit methods have been proposed.

With the recent success of differentiable geometric transformation methods [2, 31], explicit methods [5, 17, 23, 50, 57] leverage such 3D inductive biases to guide the view synthesis network to preserve 3D transformed contents, and various generative models are applied to complete the unseen regions. Explicit methods can produce high-quality novel view images in small view changes, where the content of the reference viewpoint still occupies a large portion. However, for large view changes, the image quality is degraded due to a lack of ability to generate pixels of the unseen region. To deal with this problem, outpainting with the autoregressive model is exploited to fill unseen regions [39], but generating photo-realistic images remains a challenge for explicit methods.

Submitted to 36th Conference on Neural Information Processing Systems (NeurIPS 2022). Do not distribute.

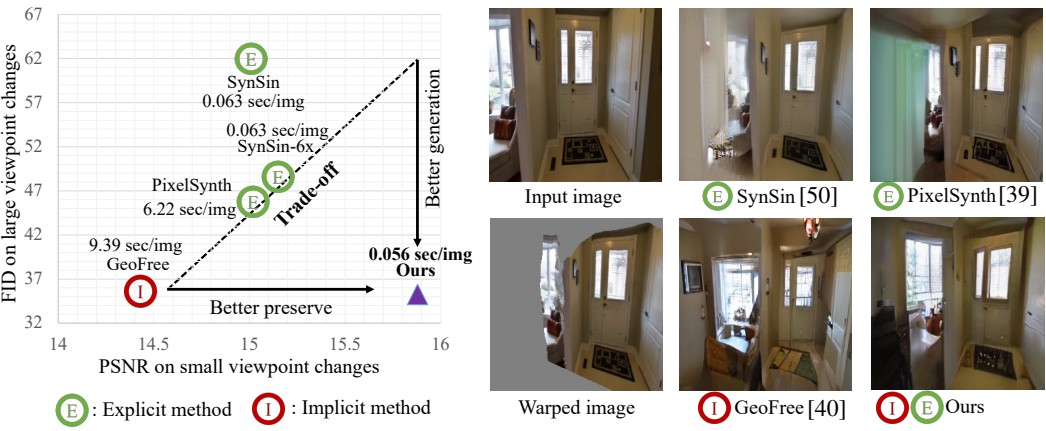

Figure 1: **Seesaw problem of explicit and implicit methods.** Explicit methods well preserve warped contents but sacrifice to fill unseen pixels (↑ PSNR on small view change, ↑ FID on large view change). Implicit methods amply fill unseen pixels but fall short of preserving seen contents (↓ PSNR on small view change, ↓ FID on large view change). Our proposed framework alleviates this seesaw problem and generates an image faster than the state-of-the-art methods.

On the other side, implicit methods [38, 40, 46] less enforce 3D inductive biases and let the model learn the required 3D geometry for view synthesis. Based on the powerful autoregressive trans-former [10], recent implicit methods learn the 3D geometry from a reference image and camera parameters. Implicitly learned 3D geometry allows the model to synthesize diverse and realistic novel view images but fails to preserve the contents of the reference image since they reduce 3D inductive biases.

To sum up, previous single-image view synthesis methods suffer from a trade-off between two objectives: 1) preserve seen contents and 2) generate semantically compatible unseen regions. Figure 1 shows an apparent trade-off that explicit methods well preserve seen contents with sacrificing the generation of unseen regions and vice versa for implicit methods. Here, we call this trade-off the *seesaw problem* and emphasize the need for combining solid points of explicit and implicit methods.

Moreover, recent methods often depend on autoregressive models, which generate individual pixels sequentially. Sequential generation causes too slower view synthesis than non-autoregressive methods, limiting their application areas, such as image animating in real-time. Therefore, we refocus on a fast and efficient non-autoregressive model for single view synthesis.

In this paper, we present a non-autoregressive framework for alleviating the seesaw problem. Our approach aims to design the architecture and loss functions. We design two parallel render blocks which explicitly or implicitly learn geometric transformations from point cloud representations. To bridge explicit and implicit transformations, we propose a novel loss function that motivates explicit features improve seen pixels of implicit features and implicit features improve unseen pixels of explicit features. Interestingly, we observe that proposed loss makes two renderers embed discriminative features and allow the model to use both renderers in a balanced way to create novel views. With the proposed architecture and the loss function, we can merge the pros of both explicit and implicit methods, alleviating the seesaw problem. As a result, our non-autoregressive framework can better preserve seen contents, better complete unseen pixels, and generate images ≈100 times faster than autoregressive methods. We validate the efficiency and effectiveness of our framework with experiments on the indoor dataset RealEstate10K [58] and the outdoor dataset ACID [23].

## 2   Related Works

**Novel view synthesis**   Given multiple images from different viewpoints of a scene, novel view synthesis aims to generate novel view images. Traditionally, multi-view geometry is utilized for synthesizing novel viewpoints [4, 6, 7, 13, 21, 42, 59]. Recently, deep neural networks have been used to rendering [15, 28, 29, 32] and several representation for view synthesis such as multi-plane image [11, 45, 58], point cloud [1], depth [44], radiance field [30, 49, 55] and voxel [25, 33, 43].

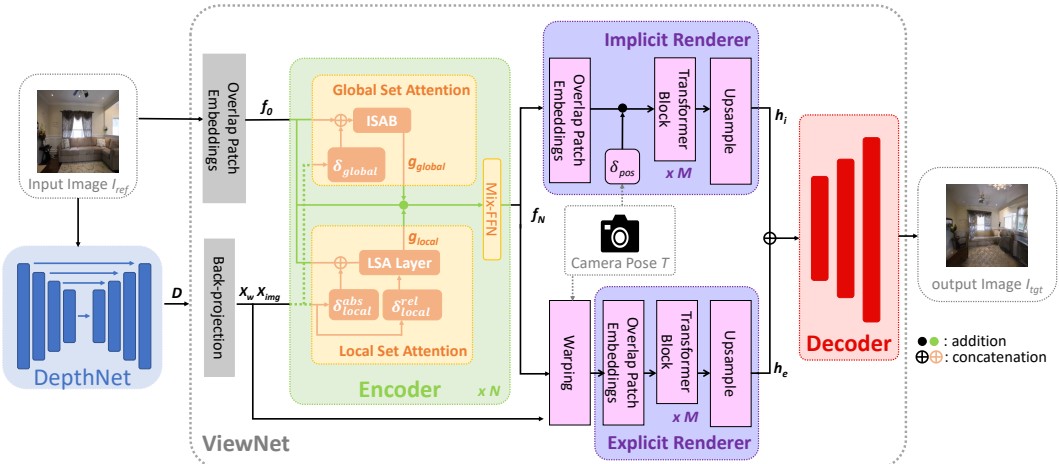

Figure 2: **An overview of network architecture.** Our network takes a reference image $I_{ref}$ and a relative camera pose $T$ as inputs. The depth estimation network (DepthNet) first predicts a depth map $D$, and the view synthesis network (ViewNet) generates a target image $I_{tgt}$ from $I_{ref}$, $D$ and $T$. Specifically, $D$ is used for calculating the 3D world coordinate $X_w$ and the normalized image coordinate $X_{img}$ at the reference viewpoint, which are passed through various positional encoding layers in the encoder (e.g., $\delta_{global}$, $\delta_{local}^{abs}$ and $\delta_{local}^{rel}$) to provide the scene structure representations. Encoded features $f_N$ are then transformed by both Implicit Renderer and Explicit Renderer with $T$. Finally, two transformed feature map, $h_i$ and $h_e$, are concatenated to generate $I_{tgt}$ by the decoder.

Single-image view synthesis is more challenging than general novel view synthesis since a single input image is only available [5, 18, 23, 38–40, 47, 50, 54]. Explicit methods directly inject 3D inductive biases into models. For example, SynSin [50] uses 3D point cloud features with estimated depth from the model, projects to novel viewpoints, and refines unseen pixels with recent generative models [3]. SynSin works well in small viewpoint changes but degrades in large viewpoint changes due to the lack of generating unseen pixels. To deal with this problem, PixelSynth [39] exploits the autoregressive outpainting model [37] with 3D point cloud representation. Despite using the slow autoregressive model, it cannot generate unseen pixels well. For an implicit method, Rombach *et al.* [40] propose a powerful autoregressive transformer. By less enforcing 3D inductive biases, this approach can generate realistic view synthesis and complete the unseen region without explicit 3D geometry. However, its inference time is long due to the autoregressive model, and it fails to preserve seen contents of a reference image. We bridge these implicit and explicit methods as a non-autoregressive architecture, which can outperform autoregressive approaches with fast inference.

**Transformer for point cloud** The transformer and self-attention have brought a breakthrough in natural language processing [8, 48] and computer vision [9]. Inspired by this success, transformer and self-attention networks have been widely applied for point cloud recognition tasks and achieved remarkable performance gain. Early methods utilize global attention for all of the point clouds, resulting in a large amount of computation and inapplicable for large-scale 3D point cloud [24, 52, 53]. Lee *et al.* [20] propose the SetTransformer module suitable for point cloud due to permutation-invariant, which uses inducing point methods and reduces computational complexity from quadratic to linear in the number of elements. Also, local attention methods is utilized to enable scalability [14, 34, 56]. Notably, among local attention methods, Fast Point Transformer [34] which uses voxel hashing-based architecture, achieves both remarkable performance and computational efficiency. Global attention may dilute important content by excessive noises as most neighbors are less relevant, and local attention may not have sufficient context due to their scope. Therefore, Our approaches use both global and local attention to deal with 3D point cloud representation.

## 3 Methodology

Given a reference image $I_{ref}$ and a relative camera pose $T$, the goal of single-image view synthesis is to create a target image $I_{tgt}$ with keeping visible contents of $I_{ref}$ and completing realistic out-of-view pixels. To achieve this, we focus on mitigating the seesaw problem between explicit and implicit

methods in terms of the network architecture and the loss function. Figure 2 describes an overview of our network architecture. The network consists of two sub-networks, the depth estimation network (**DepthNet**) and the view synthesis network (**ViewNet**). Note that the pre-trained DepthNet generates depth map $D$, which is used for ViewNet to synthesize the photo-realistic $I_{tgt}$.

## 3.1 Depth Estimation Network (DepthNet)

We train the depth estimation network for explicit 3D geometry since ground-truth depths are not available. Following Monodepth2 [12], our DepthNet is trained in a self-supervised manner from monocular video sequences. Because a ground-truth relative pose between images is available, we substitute the pose estimation network with the ground-truth relative pose. Then, we train the network on reprojection losses and smoothness losses with auto-masking in their work. After training DepthNet, we fix it during training ViewNet.

## 3.2 View Synthesis Network (ViewNet)

We design a simple view synthesis network built on architectural innovations of recent transformer models. Specifically, we exploit 3D point cloud representation to consider the relationship between the geometry-aware camera pose information and the input image.

**Encoder** The encoder aims to extract scene representations from a feature point cloud of a reference image. To deal with point clouds, we design a Global and Local Set Attention (GLSA) block which simultaneously extracts overall contexts and detailed semantics. For efficient input size of transformers, $I_{ref} \in \mathbb{R}^{H \times W \times 3}$ is encoded into $f_0 \in \mathbb{R}^{\frac{H}{4} \times \frac{W}{4} \times C}$ by an overlapping patch embedding [51], where $C$ denotes the channel dimension. Then, the homogeneous coordinates $p$ of a pixel in $f_0$ are mapped into normalized image coordinates $X_{img}$ as $X_{img}(p) = K_{\downarrow}^{-1}p$, where $K_{\downarrow}$ denotes the camera intrinsic matrix of $f_0$. Finally, 3D world coordinates of $p$ are calculated with depth map $D$ as $X_w(p) = D(p)X_{img}(p)$. Our encoder architecture is $N$ stacked GLSA block, and $i$-th GLSA block receives $f_{i-1}, X_{img}$ and $X_w$ and outputs $f_i$ with Mix-FFN [51].

*Global Set Attention.* We utilize Induced Set Attention Block (ISAB) [20] to extract global set attention between the feature point clouds. With positional encoder $\delta_{global}$ and vector concatenation operator $\oplus$, the global attention of $i$-th GLSA bock is represented as:

$$g^i_{global}(p) = ISAB(f_i(p) \oplus \delta_{global}(X_w(p))). \tag{1}$$

*Local Set Attention.* We use a modified Lightweight Self-Attention (LSA) layer [34] for the set attention in $r \times r$ local window of each pixel point. Unlike the decomposing relative position of voxels in [34], we decompose the relative position of 3D world coordinates between neighbor pixels using normalized image coordinates as:

$$X_w(p) - X_w(q) = (X_w(p) - X_{img}(p)) - (X_w(q) - X_{img}(q)) + (X_{img}(p) - X_{img}(q)), \tag{2}$$

where $q \in \mathcal{N}(p)$ is a neighbor set of homogeneous coordinates in a $r \times r$ window of $p$. With decomposition in Eq. 2, we can divide the relative positional encoding into an continuous positional encoding $\delta^{abs}_{local}$ and a discretized positional encoding $\delta^{rel}_{local}$. Then, the computation procedures for local set attention $g^i_{local}$ of $i$-th GLSA block is similar to LSA layer as:

$$\begin{aligned} l^i_{local}(p) &= f_i(p) \oplus \delta^{abs}_{local}(X_w(p) - X_{img}(p)), \\ g^i_{local}(p) &= \Sigma_{q \in \mathcal{N}(p)} S_C(\psi(l^i_{local}(p)), \delta^{rel}_{local}(X_{img}(p) - X_{img}(q)))\phi(l^i_{local}(q)), \end{aligned} \tag{3}$$

where $\psi$ and $\phi$ are MLP-layers, and $S_c(a,b) = \frac{a \cdot b}{\|a\|\|b\|}$ computes the cosine similarity between $a$ and $b$. As pixel coordinates of $p$ and $q$ are all integer, the encoding of $X_{img}(p) - X_{img}(q)$ is hashed over $r^2 - 1$ values, resulting in a space complexity reduction from $\mathcal{O}(HW \cdot r^2 \cdot C)$ to $\mathcal{O}(HW \cdot C) + \mathcal{O}(r^2 \cdot C)$.

**Rendering Module** Given the scene representations of the reference image, the rendering module learns 3D transformation from the reference viewpoint to the target viewpoint. Motivated by our observations of implicit and explicit methods, we design an Explicit Renderer(ER) and an Implicit Renderer(IR) connected in parallel to bypass the seesaw problem. The structure of the two renderers

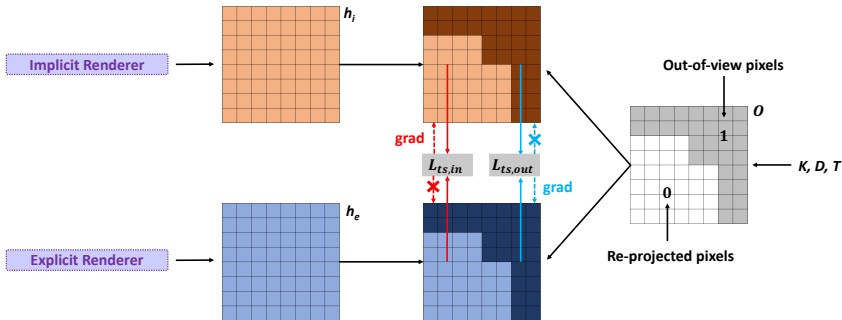

Figure 3: **An overview of our transformation similarity loss.** Two transformed features, $h_i$ and $h_e$, are complemented each other by the transformation similarity loss. Specifically, we first derive out-of-view mask **O** from $K$, $D$ and $T$. By using **O**, two transformation similarity loss, i.e., $L_{ts,in}$ and $L_{ts,out}$, are applied to encourage the discriminability of $h_i$ and $h_e$, respectively. To guide the another renderer as intended, we allow the back-propagated gradients of $L_{ts,in}$ only to the reprojected regions of $h_i$, and those of $L_{ts,out}$ only to the out-of-view regions of $h_e$.

is similar; they consist of an overlapping patch embedding, GPT architecture [36] and ResNet blocks with upsampling layers. Note that the overlapping patch embedding and upsampling layers are designed for downsampling and upsampling the input feature with the factor of 4, respectively. The major difference between the two renderers is how the relative camera pose $T$ is used for the geometric transformation.

*Explicit Renderer (ER).* Given the rotation matrix $R$ and translation vector $t$ of relative camera pose $T$, $p$ can be reprojected to the homogeneous coordinates of target viewpoint $p'$ as $p' = K_\downarrow R X_w(p) + t$. The output of encoder $f_N$ is warped by splatting operation [31] with optical flow from $p$ to $p'$. Then, warped $f_N$ goes through the explicit renderer to produce explicit feature map $h_e$.

*Implicit Renderer (IR).* Unlike the explicit renderer, the implicit renderer uses the camera parameter itself. Instead of embedding 3x4 camera extrinsic matrix, we use independent 7 parameters to embed pose information; Translation vector $t$ and axis-angle notation $(\frac{\mathbf{u}}{\|\mathbf{u}\|}, \theta)$ to parameterize rotation matrix $R$. We use a positional encoding layer $\delta_{pos}$ to embed these parameters and add them to the input of the transformer block. $f_N$ passes through the implicit renderer and outputs implicit feature map $h_i$. Please refer to the supplementary materials for details to compute the axis-angle notation.

**Decoder** Two feature maps from *ER* and *IR*, which are denoted as $h_e$ and $h_i$, are then concatenated before the decoder. We use a simple CNN-based decoder by gradually upsampling the concatenated feature map with four ResNet blocks. Instead of generating pixels in an auto-regressive manner, we directly predict all pixels in the one-path, resulting in more than 110 times faster than the state-of-the-art autoregressive methods [38–40] in generating images.

## 3.3 Loss Design for ViewNet

Following the previous single-image view synthesis methods [39, 50], we also use the $\ell_1$-loss, perceptual loss [35] and adversarial loss to learn the network. Specifically, we compute $\ell_1$-loss and perceptual loss between $I_{tgt}$ and the ground-truth image $I_{gt}$ at the target viewpoint. Also, we use the global and local discriminators [19] with a Projected GAN [41] structure and a hinge loss [22]. We observe that our methods improve the generation performance even through these simple network structural innovations. Furthermore, we introduce a transformation similarity loss $L_{ts}$ to complement two output feature maps $h_e$ and $h_i$.

**Transformation Similarity Loss** As an extension of the existing seesaw problem, $h_e$ may have better discriminability than $h_i$ in reprojected regions, conversely, $h_i$ has better delineation of out-of-view regions than $h_e$. Therefore, as shown in Fig. 3, we design the transformation similarity loss between $h_e$ and $h_i$, expecting that $h_i$ learns to keep reprojected image contests, and $h_e$ also learn to generate realistic out-of-view pixels. Specifically, we use a negative cosine similarity function $S_c$ for calculating the similarity between two feature maps, and the transformation similarity loss $L_{ts} = \lambda_{in} L_{ts,in} + \lambda_{out} L_{ts,out}$ is formulated as:

$$L_{ts,in} = -\frac{1}{\sum_p (1 - \mathbf{O}(p))} \sum_p (1 - \mathbf{O}(p)) \cdot S_c(h_i(p), detach(h_e(p))),$$

$$L_{ts,out} = -\frac{1}{\sum_p \mathbf{O}(p)} \sum_p \mathbf{O}(p) \cdot S_c(detach(h_i(p)), h_e(p)), \tag{4}$$

where $\mathbf{O}(p) \in \mathbb{R}^{\frac{H}{4} \times \frac{W}{4}}$ denotes an out-of-view mask which is derived from the depth map $D$ and the relative camera pose $T$. Note that, without detach operations, our transformation similarity loss performs the same as a simple negative cosine similarity loss between two feature maps. Thus, we detach gradients back-propagated from $L_{ts,in}$ to $h_e$ and gradients from $L_{ts,out}$ to $h_i$, because the detach operation allows the components of $L_{ts}$ to be applied to the intended area.

**Final Learning Objective**    Taken together, our ViewNet is trained on the weighted sum of a $\ell_1$-loss $L_{\ell_1}$, a perceptual loss $L_c$, an adversarial loss $L_{adv}$ and a transformation similarity loss $L_{ts}$. The total loss is then $L = L_{\ell_1} + \lambda_c L_c + \lambda_{adv} L_{adv} + L_{ts}$. We fix $\lambda_c = 1$ and $\lambda_{adv} = 0.1$ for all experiments.

Table 1: **Types of baselines and our method.** Note that InfNat [23] varies according to the number of steps, so we mark it as △.

| Types | Methods | | | | | | | |
|---|---|---|---|---|---|---|---|---|
| | Tatarchenko *et al.* [46] | Viewappearance [57] | SynSin [50] | InfNat [23] | PixelSynth [39] | GeoFree [40] | LookOutside [38] | Ours |
| Explicit | ✗ | ✓ | ✓ | ✓ | ✓ | ✗ | ✗ | ✓ |
| Implicit | ✓ | ✗ | ✗ | ✗ | ✗ | ✓ | ✓ | ✓ |
| Autoregressive | ✗ | ✗ | ✗ | △ | ✓ | ✓ | ✓ | ✗ |

# 4  Experimental Results

## 4.1  Experimental Settings

We now describe experimental settings, and please refer to the supplementary materials for further details about datasets, baselines, and our network architecture.

**Dataset**    We used two standard datasets, *RealEstate10K* [58] and *ACID* [23], which are a collection of videos mostly captured in indoor and outdoor scenes, respectively. We divided train and test sequences as in [40].

**Baselines**    To validate the effectiveness of our framework, we compared our method to previous single-image view synthesis methods : Tatarchenko *et al.* [46], Viewappearance [57], Synsin [50], InfNat [23], PixelSynth [39], GeoFree [40] and LookOutside [38]. Table 1 briefly shows whether each method is an explicit, implicit, and autoregressive model. Compared to previous methods, we use both explicit and implicit geometric transformations without an autoregressive model.

**Evaluation Details**    Because explicit and implicit methods are respectively advantageous in small view change and large view change, methods should be evaluated on several sizes of viewpoint changes for a fair comparison. Therefore, we used a ratio of out-of-view pixels over all pixels to quantify view changes, resulting in three splits are categorized into *small* (20-40%), *medium* (40-60%) and *large* (60-80%). Since evaluation datasets do not have ground-truth depth maps, we used depth maps from our pre-trained DepthNet to derive the ratio of out-of-view mask pixels. Finally, we used randomly selected 1,000 image pairs for each test split.

We use PSNR on the small split and FID [16] on the medium and large split as evaluation metrics. PSNR is a traditional metric for comparing images, which is widely used to evaluate *consistency*. Nevertheless, PSNR is a poor metric to verify the image quality on large viewpoint changes [39, 40]. Still, it can be a good metric for evaluating the preservation of reprojected pixels on small view changes. Therefore, we use PSNR on the small split to evaluate the ability to preserve seen contents. For evaluating images quality of view synthesis, FID is widely used [39, 40, 50]. Especially in the medium and large split with many out-of-view pixels, FID indicates how well the model fills out-of-view pixels and generates realistic images. We use the PSNR and FID of specific splits as evaluation metrics, but we report the PSNR and FID of all splits to show the overall trend.

Table 2: **Quantitative results on RealEstate10K and ACID.** Image quality is measured by PSNR and FID for three types of view changes, i.e., *Small*, *Medium* and *Large*. Furthermore, we show the average performance over all view changes at the end. For both datasets, best results in each metric are in **bold**, and second best are underlined.

| Dataset | Methods | Small | | Medium | | Large | | Average | |
|---|---|---|---|---|---|---|---|---|---|
| | | PSNR↑ | FID↓ | PSNR↑ | FID↓ | PSNR↑ | FID↓ | PSNR↑ | FID↓ |
| RealEstate10K [58] | Tatarchenko *et al.* [46] | 11.12 | 258.75 | 10.90 | 248.55 | 10.80 | 249.24 | 10.94 | 252.18 |
| | Viewappearance [57] | 12.51 | 142.93 | 12.79 | 110.84 | 12.44 | 147.27 | 12.58 | 133.68 |
| | SynSin [50] | 15.38 | 41.75 | 14.88 | 43.06 | 13.96 | 61.67 | 14.74 | 48.83 |
| | SynSin-6x [50] | 15.17 | 33.72 | **14.99** | 37.28 | **14.26** | 48.29 | **14.81** | 39.76 |
| | PixelSynth [39] | 14.46 | 37.23 | 13.46 | 38.39 | 12.28 | 45.44 | 13.40 | 40.35 |
| | GeoFree [40] | 14.16 | 33.48 | 13.15 | 34.21 | 12.57 | 35.28 | 13.29 | 34.32 |
| | LookOutside [38] | 12.58 | 44.87 | 12.72 | 43.17 | 12.11 | 43.22 | 12.47 | 43.75 |
| | **ours** | 15.87 | 32.42 | 14.65 | 33.04 | 13.83 | 35.26 | 14.78 | 33.57 |
| ACID [23] | Tatarchenko *et al.* [46] | 14.43 | 148.19 | 14.20 | 151.24 | 14.34 | 150.47 | 14.32 | 149.97 |
| | Viewappearance [57] | 14.46 | 161.91 | 13.58 | 203.19 | 13.21 | 218.37 | 13.75 | 194.49 |
| | SynSin [50] | 17.48 | 55.64 | 16.49 | 75.88 | **16.87** | 79.04 | **16.95** | 70.19 |
| | InfNat [23] (1-step) | 15.94 | 64.32 | 14.40 | 90.80 | 13.65 | 106.28 | 14.66 | 87.13 |
| | InfNat [23] (5-step) | 15.16 | 64.48 | 14.79 | 71.52 | 14.90 | 65.45 | 14.95 | 67.15 |
| | PixelSynth [39] | 15.81 | 53.38 | 14.33 | 63.48 | 13.53 | 65.60 | 14.56 | 60.82 |
| | GeoFree [40] | 14.80 | 53.21 | 14.24 | 58.92 | 14.22 | 54.78 | 14.42 | 55.64 |
| | **ours** | **17.52** | **42.52** | **16.54** | **51.56** | 15.81 | **49.28** | 16.62 | **47.79** |

(a) Input Image  (b) Warped Image  (c) SynSin [50]  (d) PixelSynth [39]  (e) GeoFree [40]  (f) Ours

Figure 4: **Qualitative Results on RealEstate10K and ACID.** We compare baselines to our method. The top two rows are from RealEstate10K, and the bottom two rows are from ACID.

**Implementation Details** We first resized all images into a resolution of $256 \times 256$, and normalized RGB value following [39, 50]. We trained DepthNet using a batch size 50 for $100k$ iterations and ViewNet using a batch size 32 for $150k$ iterations. Training takes about 3 days on 4 NVIDIA Geforce RTX 3090 GPUs. We used an AdamW [27] optimizer (with $\beta_1 = 0.5$ and $\beta_2 = 0.9$) and applied weight decay of 0.01. We first linearly increased the learning rate from $10^{-6}$ to $3 \cdot 10^{-4}$ during the first $1.5k$ steps, and then a cosine-decay learning rate schedule [26] was applied towards zero. In ViewNet, we used 8 GLSA blocks with local window size $r = 5$ and 6 transformer blocks in each renderer for all experiments.

## 4.2 Comparison to Baselines

We now compare our method with the state-of-the-art methods on RealEstate10K and ACID. Table 2 shows quantitative results for both datasets. The implicit method GeoFree [40] reports a lower

Table 3: **Average inference time.**

| Methods | SynSin | InfNat (5-step) | PixelSynth |
|---|---|---|---|
| Time (s/img) | 0.063 | 1.14 | 6.22 |
| Methods | GeoFree | LookOutside | Ours |
| Time (s/img) | 9.39 | 22.15 | 0.056 |

Table 4: **Ablation study on $L_{ts}$.**

| | Loss Type | | | | |
|---|---|---|---|---|---|
| | no $L_{ts}$ | $L_{ts,in}$ | $L_{ts,out}$ | $L_{ts}$(no detach) | $L_{ts}$ |
| PNSR↑ | 14.47 | 14.62 | 14.73 | 14.59 | **14.78** |
| FID↓ | 40.45 | 38.05 | 36.95 | 40.44 | **33.57** |

Table 5: **Ablation Study on the Set Attention.**

| Set Attention | | Small | | Medium | | Large | |
|---|---|---|---|---|---|---|---|
| $g_{local}$ | $g_{global}$ | PSNR↑ | FID↓ | PSNR↑ | FID↓ | PSNR↑ | FID↓ |
| ✓ | | 15.69 | 34.07 | 14.64 | 34.81 | 13.78 | 37.63 |
| | ✓ | 15.74 | 32.80 | 14.61 | 34.37 | **13.88** | 38.68 |
| ✓ | ✓ | **15.87** | **32.42** | **14.65** | **33.04** | 13.83 | **35.26** |

Table 6: **Ablation Study on hyperparameters of transformation similarity loss.**

| Loss Weight | | Small | | Medium | | Large | |
|---|---|---|---|---|---|---|---|
| $\lambda_{in}$ | $\lambda_{out}$ | PSNR↑ | FID↓ | PSNR↑ | FID↓ | PSNR↑ | FID↓ |
| 0.1 | 1 | 15.78 | 33.95 | **14.65** | 34.10 | 13.81 | 37.11 |
| 10 | 1 | 15.48 | 37.46 | 14.39 | 37.46 | 13.56 | 40.69 |
| 1 | 0.1 | 15.46 | 34.98 | 14.37 | 37.51 | 13.64 | 39.81 |
| 1 | 10 | 15.70 | 35.03 | 14.54 | 35.57 | 13.77 | 38.43 |
| 1 | 1 | **15.87** | **32.42** | **14.65** | **33.04** | **13.83** | **35.26** |

FID in the medium and large split than explicit methods such as SynSin [50] and PixelSynth [39], but its PSNR of the small split is lower. This shows that previous methods are suffered from the seesaw problem. However, our method consistently achieves the highest PSNR in the small split on both datasets, which means our method better preserves reprojected contents than previous methods. Moreover, our method also achieves the lowest FID in all splits on both datasets, and this demonstrates that our method generates better quality images with filling compatible pixels regardless of view changes. As observed in [38, 39], we note that SynSin and its variant (i.e., SynSin-6x) often produce entirely gray images, resulting they still performing competitive results in PSNR of the medium and large split. Considering this, our method stably outperforms previous methods in all splits.

Also, qualitative results in Fig. 4 illustrate that the warped regions are well-preserved and invisible parts are well-completed in our method, whereas explicit methods do not generate realistic images, and an implicit method loses the semantic information of visible contents. Specifically, GeoFree [40] does not preserve the table in the first sample and the ships floating on the sea in the third sample. Also, explicit methods [39, 50] either make the entire out-of-view regions in one color or produce a less realistic view than our method.

We confirm that mitigating the seesaw problem by well-bridged explicit and implicit geometric transformations yields high-quality view synthesis, even acquiring a generation speed of about 110 times faster than the previous autoregressive models, as shown in Table 3. The fast generation of novel view images allows our method to be scalable to various real-time applications.

### 4.3 Ablation Study: Type of Set Attention

We design the global and local set attention block to simultaneously extract overall contexts and detailed semantics. Therefore, we conducted an ablation study on RealEstate10K [58] to verify each attention improves the performance of generating novel views. Table 5 shows the quantitative result for the type of set attention. Interestingly, our local set attention improves the performance relatively in large view changes, while our global set attention performs well on small view changes. From this result, we conjecture that local and global set attention are more useful for structural reasoning of out-of-view regions and 3D scene representation of reprojected regions, respectively. Also, significant performance improvement is achieved when both attentions are used.

### 4.4 Ablation Study: Transformation Similarity Loss

The transformation similarity loss $L_{ts}$ is weighted combination of $L_{ts,in}$ and $L_{ts,out}$. To understand the effect of each component, we conducted ablation studies of transformation similarity loss on the RealEstate10K dataset. Table 4 reports the average PSNR and FID of our model by changing various components of $L_{ts}$. Results show that combining with gradient stopping operation, $L_{ts,in}$, and $L_{ts,out}$ achieves best results among the five variants. Also, either using $L_{ts,in}$ or $L_{ts,out}$ improves the performance and shows that guiding one renderer from the other renderer with the proposed loss function is effective. Notably, transformation similarity loss is not practical when the detach operation is not used. From this result, it is necessary to selectively guide unseen and seen regions by detaching the gradient.

We also performed an ablation study on balancing parameter $\lambda_{in}$ and $\lambda_{out}$. Table 6 illustrates the results varying weight of $L_{ts}$. Results show that the case of $\lambda_{in} = 1, \lambda_{out} = 1$ performs best. As mentioned above, it seems essential to complement each other in a balanced way.

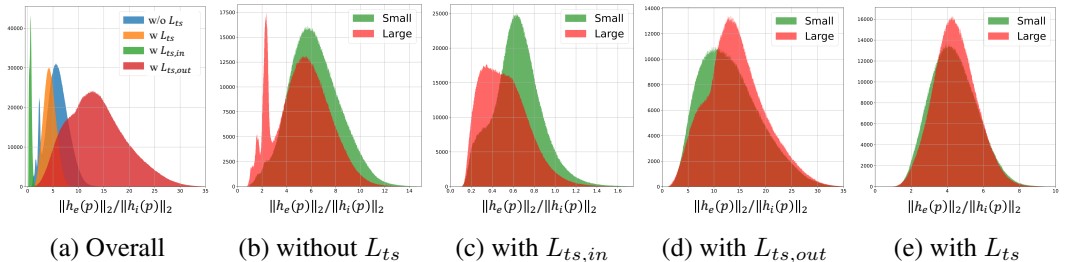

| (a) Overall | (b) without $L_{ts}$ | (c) with $L_{ts,in}$ | (d) with $L_{ts,out}$ | (e) with $L_{ts}$ |

Figure 5: **Histogram of** $||h_e(p)||_2/||h_i(p)||_2$ **on the small and large split of RealEstate10K dataset.**

### 4.5 Dependency Analysis between Implicit and Explicit Renderers

Our proposed architecture exploits the implicit and explicit renderer and mixes their outputs for decoding view synthesis results. To understand the dependency between two renderers, we analyze the norm of output feature maps. For a spatial position $p$, the norm ratio of two spatial features $||h_e(p)||_2/||h_i(p)||_2$ can represent how much depends on the explicit feature $h_e(p)$ compared to implicit feature $h_i(p)$. For example, if the ratio is large, the model depends on the explicit renderer than the implicit renderer at position $p$. We compare histograms of the norm ratio by changing the components of $L_{ts}$ and data splits as shown in Fig. 5.

Figure 5a depicts that using $L_{ts,out}$ and $L_{ts,in}$ tends to make the model more dependent on explicit and implicit features, respectively, compared to our method trained without $L_{ts}$. Furthermore, these tendencies are more apparent in difficult cases (i.e., large split) as shown in Fig. 5c–5d. From our observations, we conjecture that guiding only a specific renderer improves the discriminability of that renderer, resulting in the model depending on the improved renderer. Surprisingly, the model trained on combining all components of $L_{ts}$ uses both renderers in a balanced way, and there is less bias in norm ratio even according to data splits as shown in Fig. 5e.

The effectiveness of our transformation similarity loss is confirmed by comparing it to our method that is trained without $L_{ts}$. Figure 5b shows that our model trained without $L_{ts}$ has some outliers for large view changes despite there being less bias according to data splits. We observe these outliers are derived when the model fails to generate realistic out-of-view regions, especially in challenging settings, such as the network having to create novel views for both indoor and outdoor scenes, as shown in Fig. 6. We also confirm that our model trained with $L_{ts}$ performs well even in extreme cases, informing that $L_{ts}$ improves two renderers to embed discriminative features. Collectively, $L_{ts}$ improves the discriminability of output features from two renderers and makes the behavior of the model stable, resulting in alleviating the seesaw problem.

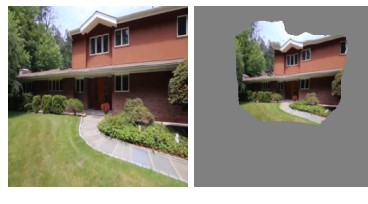

(a) Input Image  (b) Warp Image

(c) Without $L_{ts}$  (d) With $L_{ts}$

Figure 6: **Visual ablation study.** Without the transformation similarity loss, our model complete textured out-of-view regions but not realistic enough than our model trained with the transformation similarity loss.

## 5 Conclusion

We have introduced a single-image view synthesis framework by bridging explicit and implicit renderers. Despite using autoregressive models, previous methods still suffer from the seesaw problem since they use only one explicit or implicit geometric transformation. Thus, we design two parallel renderers to mitigate the problem and complement renderers with transformation similarity loss. Alleviating the seesaw problem allows the network to generate novel view images better than previous methods, even with a non-autoregressive structure. We note that the effectiveness of bridging two renderers can be applied in other tasks, such as extrapolation. We believe that our work can prompt refocusing on non-autoregressive architecture for single-image view synthesis.

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
