# 1 Supplemental Materials

# 2 Contents

# A  Notations

Images and Feature maps:

| | |
|---|---|
| $I_{ref}$ | Input image |
| $I_{tgt}$ | Generated target image |
| $I_{gt}$ | Ground-truth target image |
| $D$ | The estimated Depth Map from DepthNet |
| $f_i$ | The $i$-th output point feature of the encoder |
| $l_{local}^i$ | The continuous positional encoded feature of the $i$-th LSA layer |
| $g_{global}^i$ | The $i$-th global set attention of the encoder |
| $g_{local}^i$ | The $i$-th local set attention of the encoder |
| $h_i$ | The output feature map of the implicit renderer |
| $h_e$ | The output feature map of the explicit renderer |
| $\mathbf{O}$ | The out-of-view mask |

Camera parameters and Coordinates:

| | |
|---|---|
| $K$ | Input camera intrinsic matrix for a resolution of $H \times W$ |
| $T$ | Input relative camera pose matrix |
| $R$ | The rotation matrix of $T$ |
| $t$ | The translation vector of $T$ |
| $\mathbf{u}/\|\mathbf{u}\|$ | The normalized axis that is not changed by $R$ |
| $\theta$ | The amount of rotated angle of $R$ |
| $X_{img}$ | A set of normalized image coordinates |
| $X_w$ | A set of 3D world coordinates |
| $\mathcal{N}(p)$ | A set of neighbor homogeneous coordinates of $p$ |

MLP-layers and Operations:

| | |
|---|---|
| $\delta_{global}$ | A position encoding layer in ISAB |
| $\delta_{local}^{abs}$ | A continuous position encoding layer in LSA layer |
| $\delta_{local}^{rel}$ | A discretized position encoding layer in LSA layer |
| $\psi$ | A query projection layer in LSA layer |
| $\phi$ | A value projection layer in LSA layer |
| $\delta_{pos}$ | A positional encoding layer for camera parameters |
| $\oplus$ | A vector concatenation operation |
| $S_c(\cdot)$ | A cosine similarity operation |

# B  Experimental Details

Our code is available at `https://anonymous.4open.science/r/Bridging_Implicit_Explicit_viewsyn-1322/README.md`.

## B.1  Datasets

To select training image pairs from video clips in RealEstaet10K [15] and ACID [3], our selection protocol proceeds similarly to the previous work [11]. However, we experimentally set selection limits that allow the network to learn both small and large view changes and exclude situations of entering different rooms. Specifically, we set the range of angle ($^\circ$), translation ($m$) and frame differences (frames) to $[10, 60]$, $[0, 3]$ and $[0, 100]$ for both datasets, respectively.

## B.2  Baselines

**SynSin [11]**    SynSin [11] uses a point cloud representation for single-image view synthesis. Similar to our method, it does not require any ground-truth 3D information and uses a differentiable point cloud renderer. The point cloud representation projected by the renderer is refined to generate novel view images. Since the official code is publicly available, we use it for implementation [1]. SynSin-6x, which is a variant of SynSin trained on large viewpoint changes, is introduced in [7]. For implementation of SynSin-6x, we adopt the official code of PixelSynth [7] [2].

**PixelSynth [7]**    SynSin achieves remarkable view synthesis results in small viewpoint changes, but it fails to fill the unseen region of novel view images realistically. PixelSynth utilizes the outpainting strategy for supplementing the ability to complete the unseen region of SynSin. Although a slow autoregressive model is used for outpainting, PixelSynth still performs poorly in filling the out-of-view pixels. The official code is publicly available, and we utilize it for implementation [2].

**GeoFree [8]**    With the powerful transformer and autoregressive model, GeoFree [8] shows that the model can learn the 3D transformation needed for the single-image view synthesis. Its view synthesis results are realistic, but it fails to maintain the seen contents. We adopt the official code for implementation [3].

**Tatarchenko *et al.* [10]**    Tatarchenko *et al.* [10] use a convolutional neural network to predict an RGB image and a depth map for arbitrary viewpoint. We adopt the implementation of SynSin [11] [1].

**Viewappearance [14]**    Viewappearance [14] predicts the flow and warps the reference image to the target view with this flow. For implementation, we used the implementation of SynSin [11] [1].

**InfNat [3]**    Infinite Nature [3] focuses on nature scenes and generates a video from an image and a camera trajectory. InfNat uses a pretrained MiDAS [5] to estimate depth maps, and novel views are generated based on explicit geometric transformations. We evaluate the performance for 1-step (i.e., direct generation) and 5-step (i.e., gradual generation for target view). We adopt the official code for implementation [4].

**LookOutside [6]**    Ren *et al.* [6] focus on long-term view synthesis with the autoregressive model. Novel views are generated time-sequentially, which takes more generation time than GeoFree [8]. LookOutside utilizes a pretrained encoder-decoder in GeoFree [8] for mapping the images to tokens. We adopt the official code for implementation [5].

---

[1] `https://github.com/facebookresearch/synsin`

[2] `https://github.com/crockwell/pixelsynth`

[3] `https://github.com/CompVis/geometry-free-view-synthesis`

[4] `https://github.com/google-research/google-research/tree/master/infinite_nature`

[5] `https://github.com/xrenaa/Look-Outside-Room`

## B.3 Architectural Details

**Encoder** The channel dimension $C$ of $f_0$ is set to 256, and all positional encoding layers embed into 32 channels. Thus, we first apply MLP-layers to embed $C$-dimensional input features for ISAB and LSA layers, where each MLP-layer takes $(C + 32)$-dimensional features and outputs $C$-dimensional features. For a global set attention block, we first define a MAB (Multihead Attention Block) as:

$$
\begin{aligned}
\text{Attention}(Q, K, V) &= \text{Softmax}(\frac{QK^T}{\sqrt{d_{head}}})V, \\
H &= \text{LayerNorm}(X + \text{Attention}(X, Y, Y)), \\
\text{MAB}(X, Y) &= \text{LayerNorm}(H + rFF(H)),
\end{aligned}
\tag{1}
$$

where rFF denotes any row-wise feed-forward layer, and we use the same rFF in [2]. Then, using two MABs and $m$ inducing points $I \in \mathbb{R}^{m \times C}$, we define the global set attention for $n$ points as:

$$
\begin{aligned}
\text{ISAB}_m(X) &= \text{MAB}(X, G) \in \mathbb{R}^{n \times C}, \\
\text{where} \quad G &= \text{MAB}(I, X) \in \mathbb{R}^{m \times C}.
\end{aligned}
\tag{2}
$$

Note that, we compute the global set attention for $n = \frac{H}{4} \cdot \frac{W}{4}$ points, and fix $m = 32$. Moreover, in the LSA layer, we fix local window size $r = 5$ considering the previous point transformer networks where Point Transformer [13] uses 32 neighbors, and Fast Point Transformer [4] set local window size as 3 or 5. Finally, we apply Mix-FFN [12] to extract the $i$-th output point feature of the encoder $f_i$ as:

$$
\begin{aligned}
f_i &= \text{Mix-FFN}(X_i) = \text{MLP}(\text{GELU}(\text{CONV}_{3\times3}(\text{MLP}(X_i)))) + X_i, \\
\text{where} \quad X_i &= f_{i-1} + g_{global}^i + g_{local}^i.
\end{aligned}
\tag{3}
$$

**Rendering Module** We first illustrate the axis-angle notation, which is used for the implicit renderer. Axis-angle notation consists of *normalized axis*, i.e., a normalized vector along the axis is not changed by the rotation, and *angle*, i.e., the amount of rotation about that axis. We use a standard method that defines the eigenvector **u** of the rotation matrix by using the property that $R - R^T$ is a skew-symmetric matrix as:

$$
[\mathbf{u}]_X \equiv (R - R^T), \ i.e., \ \mathbf{u} = [r_{32} - r_{23}, \ r_{13} - r_{31}, \ r_{21} - r_{12}]^T,
\tag{4}
$$

where $r_{ij}$ is the element of $R$ located at the $i$-th row and the $j$-th column. We can also calculate the rotation angle $\theta$ from the relationship between the norm of eigenvector $\|\mathbf{u}\|$ and the trace of the rotation matrix $tr(R)$. Following the existing theorem [1, 9], the rotation angle $\theta$ is derived as:

$$
\theta = \arctan\left(\frac{\|\mathbf{u}\|}{tr(R) - 1}\right).
\tag{5}
$$

This notation often fails when the camera rotates near $180°$; however, we do not cover such an extreme movement of the camera. With a translation vector $t$, seven pose parameters (i.e., $(\frac{\mathbf{u}}{\|\mathbf{u}\|}, \theta, t)$) are processed into $\delta_{pos}$, and then added to all output tokens of the overlapping patch embedding layer. Also, for both renderers, we use the $\text{MAB}(Z, Z)$ described in Eq. 1 as transformer blocks for input feature $Z$, with MiX-FFN [12] as the feed-forward layer.

## C Additional Results

### C.1 Quantitative Results

**PSNR measured for reprojected regions.** To clarify the performance of preserving seen contents, we evaluate the PSNR only for reprojected pixels; the metric is denoted as *PSNR-vis*. Table 1 and Table 2 show the PSNR-vis for RealEstate10K [15] and ACID [3], respectively. Recent explicit methods [3, 7, 11] perform better than recent implicit methods [6, 8], which confirms that explicit methods better preserve the seen contents than implicit methods. Note that our method consistently achieves the highest PSNR-vis for all splits, outperforming previous methods by a large margin.

Table 1: **PSNR-vis on RealEstate10K [15].**

| Methods | PSNR-vis↑ | | | |
|---|---|---|---|---|
| | Small | Medium | Large | Average |
| Tatarchenko et al. [10] | 11.16 | 10.75 | 10.70 | 10.87 |
| Viewappearance [14] | 12.39 | 12.89 | 12.50 | 12.59 |
| SynSin [11] | 15.67 | 15.46 | 14.72 | 15.28 |
| SynSin-6x [11] | 15.43 | 15.54 | 14.92 | 15.30 |
| PixelSynth [7] | 15.62 | 15.60 | 14.64 | 15.29 |
| GeoFree [8] | 14.89 | 14.37 | 13.60 | 14.29 |
| LookOutside [6] | 12.78 | 13.13 | 12.54 | 12.82 |
| **ours** | **16.94** | **15.97** | **15.36** | **16.09** |

Table 2: **PSNR-vis on ACID [3].**

| Methods | PSNR-vis↑ | | | |
|---|---|---|---|---|
| | Small | Medium | Large | Average |
| Tatarchenko et al. [10] | 14.53 | 14.34 | 14.62 | 14.50 |
| Viewappearance [14] | 14.66 | 13.76 | 13.22 | 13.88 |
| SynSin [11] | 18.05 | 17.16 | 17.32 | 17.51 |
| InfNat [3] (1-step) | 16.97 | 15.74 | 15.24 | 15.98 |
| InfNat [3] (5-step) | 15.76 | 15.44 | 15.62 | 15.61 |
| PixelSynth [7] | 17.61 | 16.22 | 15.32 | 16.38 |
| GeoFree [8] | 15.26 | 14.86 | 14.67 | 14.93 |
| **ours** | **18.17** | **17.58** | **17.88** | **17.88** |

**More Explorations of the Transformation Similarity Loss** As we consistently mention the balance of the two renderers, we further explore the case where the norms of $h_e$ and $h_i$ are the same. Consequently, we use a $\ell_1$-loss instead of the negative cosine similarity loss to strengthen the coupling between the implicit renderer and the explicit renderer. Table 3 shows that tight bridging between two renderers degrades the generation power. Since the two renderers learn the different 3D scene representations for novel view synthesis, constraining $h_i$ and $h_e$ exactly the same causes a conflict in learning representations.

We also analyze the effect of the transformation similarity loss compared to using the out-of-view mask as an additional input for the decoder. If the out-of-view mask **O** is concatenated with $h_i$ and $h_e$, the decoder can learn to fuse the rendered feature $h_i$ and $h_e$ without our transformation similarity loss. As shown in Table 4, additional mask information achieves slight improvements for PSNR-vis, but the improvements in FID are negligible considering that it takes up a little more memory. Note that two renderers without our transformation similarity loss do not sufficiently represent semantic information, although additional mask information is used. On the other side, our method achieves significant performance improvement in both metrics while using the same memory as our method trained without $L_{ts}$.

Table 3: **Ablation Study on the Similarity Operation in $L_{ts}$.** PSNRs and FID are measured on RealEstate10K [15]. Note that the strict coupling between $h_i$ and $h_e$ reduces the generation performance in both PSNR and FID.

| Operation Type | Small | | | Medium | | | Large | | |
|---|---|---|---|---|---|---|---|---|---|
| | PSNR-vis↑ | PSNR-all↑ | FID↓ | PSNR-vis↑ | PSNR-all↑ | FID↓ | PSNR-vis↑ | PSNR-all↑ | FID↓ |
| $\ell_1$-loss | 16.43 | 15.46 | 42.21 | 15.66 | 14.47 | 44.97 | 15.11 | 13.72 | 55.18 |
| $-S_c(\cdot)$ | **16.94** | **15.87** | **32.42** | **15.97** | **14.65** | **33.04** | **15.36** | **13.83** | **35.26** |

Table 4: **Effects of the transformation similarity loss.** PSNRs and FID are measured on RealEstate10K [15]. Our transformation similarity loss is more effective than just using the out-of-view mask as an additional input of the decoder.

| Operation Type | Small | | | Medium | | | Large | | |
|---|---|---|---|---|---|---|---|---|---|
| | PSNR-vis↑ | PSNR-all↑ | FID↓ | PSNR-vis↑ | PSNR-all↑ | FID↓ | PSNR-vis↑ | PSNR-all↑ | FID↓ |
| No $L_{ts}$ | 16.55 | 15.41 | 35.52 | 15.86 | 14.42 | 38.10 | 15.30 | 13.57 | 47.74 |
| **O**$(p)$ as feature | 16.86 | 15.23 | 34.74 | 15.92 | 14.51 | 36.10 | **15.36** | 13.31 | 46.43 |
| ours | **16.94** | **15.87** | **32.42** | **15.97** | **14.65** | **33.04** | **15.36** | **13.83** | **35.26** |

**Effects of the Adversarial Loss** Since we use a different adversarial loss compared to SynSin [11], we further conducted an ablation study on the effect of the adversarial loss. Table 5 shows our adversarial loss improves the generation power of SynSin, but it is still a worse FID score than our method. We confirm that our method is not just boosted with a more powerful adversarial loss. Our architecture advances bridging explicit and implicit geometric transformations with transformation similarity loss contributes significantly to performance gain.

Also, the new GAN loss does not solve the seesaw problem as it improves SynSin in FID by sacrificing PSNR-vis. Explicit methods still have room for improvement in completing out-of-view regions, but more advanced generative models cannot solve the seesaw problem. Note that our bridging scheme and the transformation similarity loss are necessary to mitigate the seesaw problem.

Table 5: **Effects of the adversarial loss.** PSNRs and FID are measured on RealEstate10K [15].

| Operation Type | Small | | | Medium | | | Large | | |
|---|---|---|---|---|---|---|---|---|---|
| | PSNR-vis↑ | PSNR-all↑ | FID↓ | PSNR-vis↑ | PSNR-all↑ | FID↓ | PSNR-vis↑ | PSNR-all↑ | FID↓ |
| SynSin | 15.67 | 15.38 | 41.75 | 15.46 | **14.88** | 43.06 | 14.72 | 13.96 | 61.67 |
| SynSin + our $L_{adv}$ | 15.45 | 15.23 | 40.43 | 15.31 | **14.88** | 39.13 | 14.51 | **13.98** | 54.27 |
| ours | **16.94** | **15.87** | **32.42** | **15.97** | 14.65 | **33.04** | **15.36** | 13.83 | **35.26** |

## C.2 Qualitative Results

We further evaluate our method on different sizes of viewpoint changes as shown in Fig. 1 and Fig. 2. We also visualize additional qualitative results in Fig. 3. Note that our method synthesizes novel views consistent with $I_{ref}$ and realistic out-of-view regions, regardless of the view change.

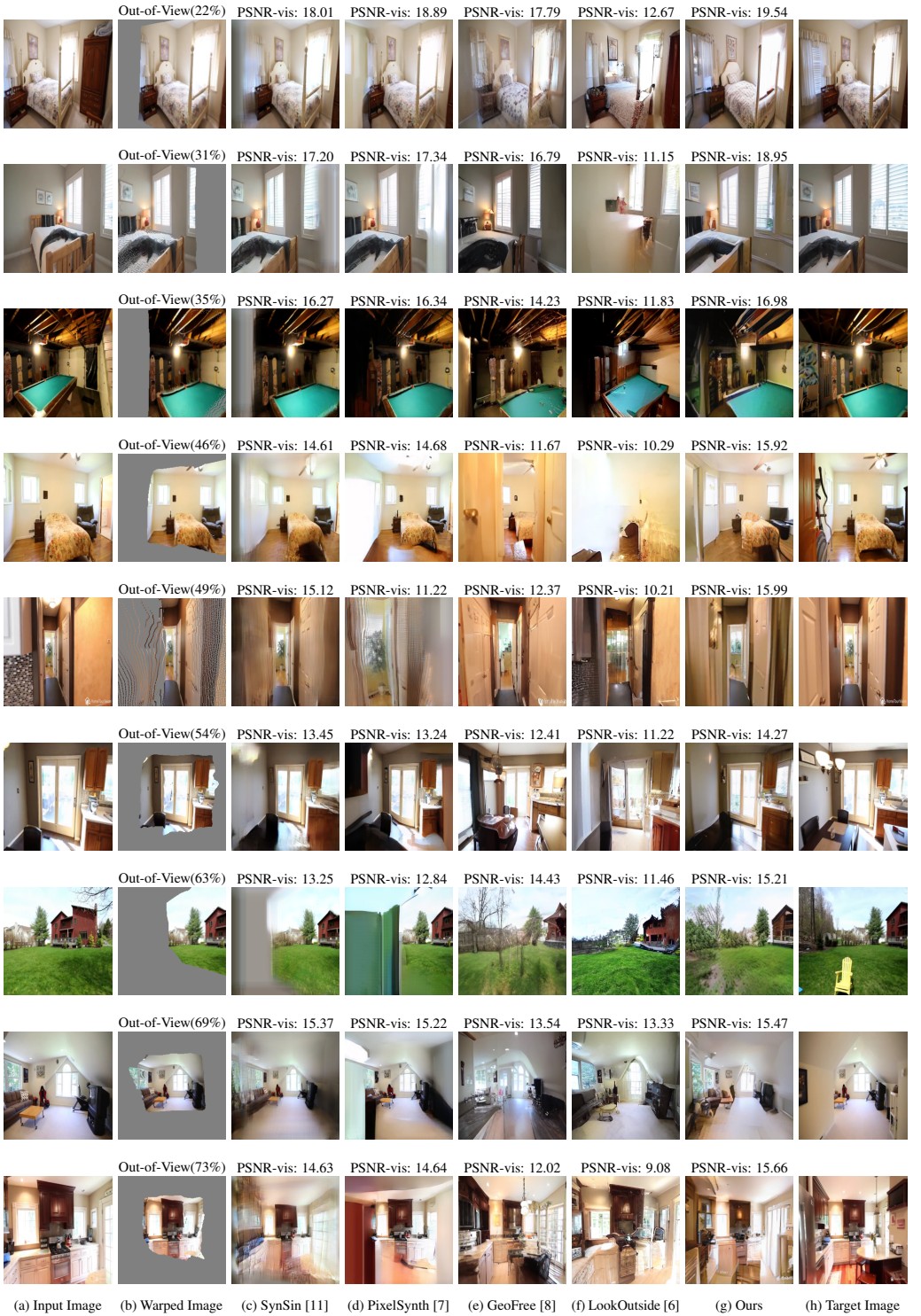

Figure 1: **Qualitative Results on RealEstate10K [15].**

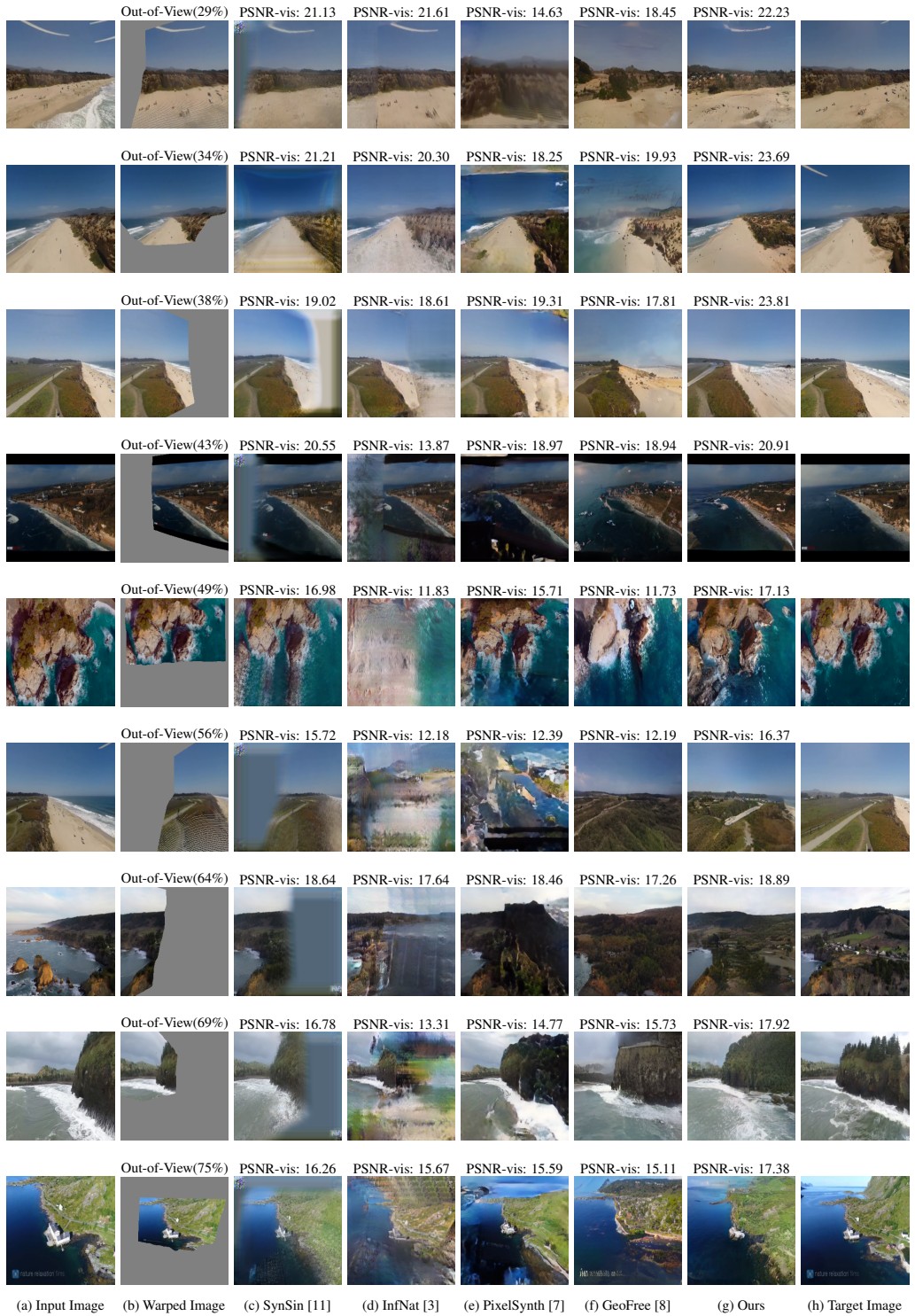

Figure 2: **Qualitative Results on ACID [3].** For InfNat [3], we report examples with higher PSNR-vis scores in either 1-step or 5-step variants.

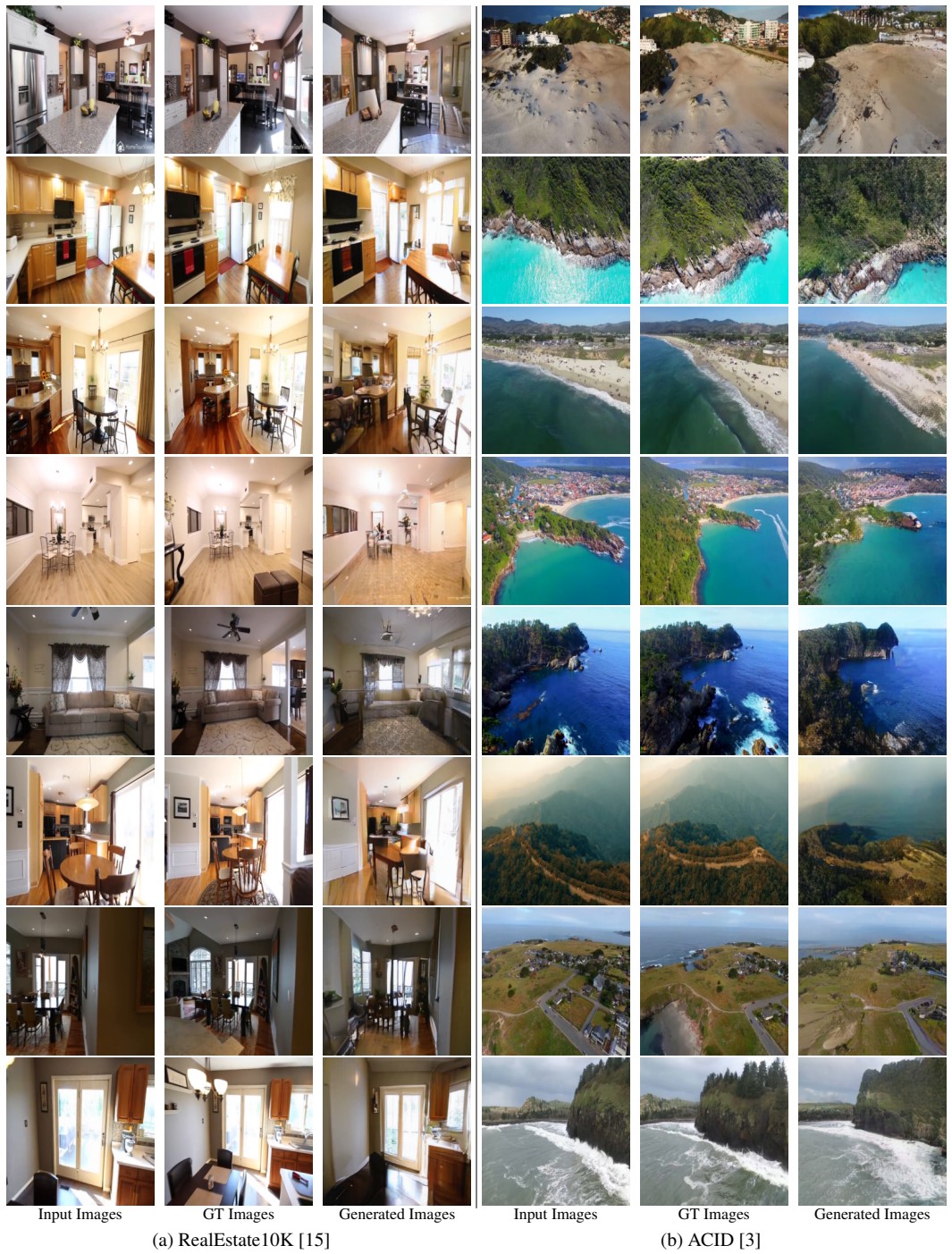

| Input Images | GT Images | Generated Images | Input Images | GT Images | Generated Images |

(a) RealEstate10K [15]                    (b) ACID [3]

Figure 3: **Additional Qualitative Results.**

## D Discussion

**Failure Cases**  Since we train the depth estimation network in a self-supervised manner, some reprojected regions can be mismatched with the target image due to various reasons (e.g., occlusion and textureless regions), reducing the accuracy of explicitly rendered features. Most mismatches are corrected by balancing with the implicit renderer, but occlusions in textureless regions may create some artifacts in the generated image.

**Limitations and Future Works**  As many possible target images can be consistent with the reference image and the relative camera pose, a probabilistic framework may generate better novel views than deterministic models. We will explore how to combine our bridging scheme and recent probabilistic frameworks in future work.

**Potential Social Negative Impact**  Moving the camera from a photograph with single-image view synthesis can be used to affect privacy adversely. As the model trained on specific data can be biased, training data must be carefully selected.