# OpenReview forum: "Bridging Implicit and Explicit Geometric Transformations for Single-Image View Synthesis"
_NeurIPS.cc/2022/Conference — NeurIPS 2022 Submitted_

### Official Review · Reviewer_gwT5 · 2022-07-04

**Rating:** 6
**Confidence:** 4
**Soundness:** 3 good
**Presentation:** 3 good
**Contribution:** 3 good

**Summary:**

The paper proposes a non-autoregressive architecture for single-image view synthesis that combines both explicit and implicit methods. Two parallel renders extract explicit feature maps and implicit feature maps correspondingly, and the proposed transformation similarity loss encourages the consistency between two feature maps in the reprojected region and the out-of-view region.

**Questions:**

- In Figure 4, there are check-board artifacts for your method (e.g. the 2nd row), while the autoregressive methods don’t have that. Do you think it can be caused by the upsampling in the decoder?


**Limitations:**

The limitations are discussed in Sec 5.

**Strengths And Weaknesses:**

Originality: The idea of combining the explicit and implicit methods for single-image view synthesis is interesting. Also, the non-autoregressive framework reduces the inference time significantly. Some existing methods like GeoFree[40] also compare both explicit and implicit geometric transformation, but they didn’t combine the advantages of explicit and implicit methods, so the paper is sufficiently novel.

Quality: Overall the proposed method is well-described.
- As mentioned in the paper, the PSNR is a good metric for evaluating the preservation of reprojected pixels on small changes only. Is it possible to evaluate it for reprojected regions only? In table 2, the Synsin[50] has very competitive PSNR performance (even better for both medium and large split); in Figure 4, the Synsin also shows a better ability to preserve projected contents (e.g. the carpet near the door in 1st row and the droplight in 2nd row), so the explanation in Line 230-232 is not very convincing to me.
- Maybe it is worth moving the ablation study on the Type of Set Attention in supp. to the main paper since the design of the encoder didn't be discussed in the experiments section, though it seems a novel contribution.

Clarity:  The paper is well-written and the motivation is clear. Some parts of the experiments can be explained better:
- The definition of the explicit methods is not clear to me. For example, the Tatarchenko [46] is defined as an explicit method, but the way they parametrize the pose is very similar to the proposed Implicit renderer (angles only, no explicit warping introduced to the features to generate the image).
- It’s good to evaluate methods on different sizes of viewpoint changes. Is it possible to also show the qualitative results respectively?  Currently, Figure 4 is not clear since it only has the input source image and doesn’t show the target image, so it’s difficult to know if they are examples of small or large viewpoint changes.

Significance: The idea of rendering two feature maps in parallel can be general and it is possible to extend it to other frameworks.

---

> ### Author Response · Authors · 2022-08-02
> **Response to gwT5**
>
> Dear Reviewer gwT5,
>
> We are very grateful for the positive and thoughtful review by Reviewer gwT5, _e.g., The paper is sufficiently novel, overall the proposed method is well-described, the paper is well-written, the motivation is clear, the idea of rendering two feature maps in parallel can be general, and it is possible to extend it to other frameworks_. We found them very helpful in improving our manuscript. We will address all concerns raised by the reviewer and revise the paper accordingly.
> ***
> ### Evaluate PSNR for only reprojected regions.
> For RealEstate10K and ACID dataset, the ground-truth depth map is not available. Therefore, we can not accurately calculate observed and unobserved regions, and we did not report PSNR on observed regions for rigorous experiments. This can induce the question of whether reprojected regions do not seem to be preserved well.
>
> With the trained Monodepth2 model, we report PSNR on reprojected regions calculated by estimated depth maps. The results are shown in Table below. By comparing PSNR on seen pixels to baselines, our method consistently outperforms them, demonstrating better content preservation in observed regions. These results are different from those in Table 2, which show that when PSNR is calculated for the entire image, our method performs marginally worse than SynSin in large splits. We can demonstrate that our method effectively preserves seen pixels by measuring PSNR on seen pixels. We are grateful for the reviewer's suggestion and have updated supplementary material for Appendix C.1 to include the specifics and results.
> |       RealEstate10K        |||| | ACID|||||
> |-|-|-|-|-|-|-|-|-|-|
> | Methods         |   Small   |   Medium  |   Large   | Average | Methods                 |   Small   |   Medium  |   Large   |  Average  |
> | Tartarchenko    |   11.16   |   10.75   |    10.70   |  10.87  | Tatarchenko             |   14.53   |   14.34   |   14.62   |    14.50   |
> | Viewappearance  |   12.39   |   12.89   |    12.50   |  12.59  | Viewappearance          |   14.66   |   13.76   |   13.22   |   13.88   |
> | SynSin|   15.67   |   15.46   |   14.72   |  15.28  | SynSin                  |   18.05   |   17.16   |   17.32   |   17.51   |
> | SynSin-6x|   15.43   |   15.54   |   14.92   |   15.30  | InfiniteNature (1-step) |   16.97   |   15.74   |   15.24   |   15.98   |
> | PixelSynth|   15.62   |    15.60   |   14.64   |  15.29  | InfiniteNature (5-step) |   15.76   |   15.44   |   15.62   |   15.61   |
> | GeoFree|   14.89   |   14.37   |    13.60   |  14.29  | PixelSynth              |   17.61   |   16.22   |   15.32   |   16.38   |
> | LookOutSideRoom |     12.78    |     13.13     |     12.54     |    12.82    | GeoFree                 |   15.26   |   14.86   |   14.67   |   14.93   |
> | Ours| **16.94** | **15.97** | **15.36** |  **16.09**  | Ours| **18.17** | **17.58** | **17.88** | **17.88** |
> ***
> ### Additional qualitative results on different sizes of viewpoint changes.
> We agree with the reviewer’s suggestion that qualitative results with various sizes of viewpoint changes are helpful in improving our manuscript. We also agree that adding the target image can directly represent the size of viewpoint changes. We report qualitative results with the out-of-view ratio, reprojected image, target image and PSNR measured on seen regions which are obtained from trained Monodepth2. This makes the comparison easy according to viewpoint size differences and shows the effectiveness of our method. We appreciate the valuable suggestion and add results in Appendix C.2 in the revised supplementary material.
> ***
> ### Check-board artifacts.
> Check-board artifacts are analyzed in [A]. We knew about it and considered it for designing our decoder. Qualitative results in the 2nd row of Fig. 4 may appear that there are checkerboard artifacts, and we observe that these artifacts rarely occur in incorrectly reprojected regions. We postulate that artifacts appear when the fixing process fails internally because the model learns to correct incorrectly reprojected regions by using implicit branches during training. We add the discussion in Appendix D of the revised supplementary material. Nevertheless, as shown in our additional qualitative results in Appendix C.2 of the revised supplementary material clearly show that our method performs well.
>
> ***
> ### Move the ablation on the type of set attention to the main paper.
> We sincerely appreciate the reviewers' recommendations. We moved the ablation study on the type of set attention to the revised main paper. Because of limited space, we move the limitation and potential social negative impact to the revised supplementary material.
> ***
> ### Definition of the explicit methods.
> We apologize for the confusion caused by our mistake. Tartarchenko _et al._ propose an implicit method and we correct it.
>
> ***
> ### References
> [A] Odena, Augustus, Vincent Dumoulin, and Chris Olah. "Deconvolution and checkerboard artifacts." Distill 1.10 (2016): e3..

---

> > ### Comment · Reviewer_gwT5 · 2022-08-09
> > **Response to the authors**
> >
> > Thank you for your detailed clarifications. The new evaluation results look good and my main concern has been resolved. I will keep my recommendation as weak accept.

---

> > > ### Author Response · Authors · 2022-08-09
> > > **Response to Reviewer gwT5**
> > >
> > > We thank Reviewer gwT5 for the positive comments and constructive feedback. Thank you for taking the time to respond to the rebuttal.

---

### Official Review · Reviewer_qWX6 · 2022-07-07

**Rating:** 6
**Confidence:** 3
**Soundness:** 3 good
**Presentation:** 3 good
**Contribution:** 3 good

**Summary:**

This paper proposes a new method for single-image view synthesis. The authors point out that existing methods are faced with a "seesaw" problem, i.e., a trade-off between preserving observed contents and synthesizing out-of-view regions. To address this issue, the authors propose a method that shares the advantages of both explicit and implicit geometric transformations, where the former is good at preserving existing contents while the latter is good at synthesizing out-of-view contents. A transformation similarity loss is proposed, which motivates the explicitly rendered feature and implicitly rendered feature to learn the best from each other. The pipeline is a feed-forward model without an autoregressive module, thus significantly faster than other autoregressive models. Experiments on two datasets show that the proposed method achieves obvious improvements over other implicit or explicit baselines both qualitatively and quantitatively. Ablation studies are carefully conducted to verify the effectiveness of the proposed transformation similarity loss and the dependency between explicit and implicit renderers.

**Questions:**

1. I suggest adding InfiniteNature (or a warp+inpainting method) as a baseline if possible.

2. I suggest manually fusing the two features according to the out-of-view mask or sending the mask as an additional input to the final decoder so that the decoder can learn to fuse them.

3. Authors may consider adding some video results, which would be more intuitive for view synthesis tasks.

4. Authors may comment on any weakness listed above if they disagree.

**Limitations:**

It seems that for reprojected regions, some mismatch to the input image can still be observed in the generated results. Thus, the "preserving reprojected contents" objective is still not fully fulfilled and has room for improvement.

**Strengths And Weaknesses:**

Strengths:

1. This paper points out the trade-off between explicit and implicit geometric transformation-based methods, and proposes a hybrid model that integrates the best of both, which is new and reasonable. The synergy effects of explicit and implicit renderers are well demonstrated in experiments.

2. The proposed transformation similarity loss is novel to me. Its effectiveness is clearly verified in Tab.4 and Fig.5, which shows that it helps the explicitly rendered feature and implicitly rendered feature to benefit from each other.

3. The proposed method shows clear improvements over other baseline approaches in quantitative and qualitative evaluations. Results in Fig.4 show that it better preserves existing contents and synthesizes out-of-view contents.


Weaknesses:

1. I feel that the "seesaw" problem is somewhat exaggerated. In my humble opinion, a simpler and more straightforward way to solve this problem is to reproject or warp existing pixels to the target view based on the depth, and then use a generative model to synthesize the out-of-view pixels conditioned on the warped pixels, like inpainting. In this way, it naturally preserve existing contents and synthesize out-of-view contents. InfiniteNature [23] is designed in this way but is not compared. If this design can address the "seesaw" problem, then the contribution of this work is not strong. Otherwise, the authors should add some discussion to explain why.

2. Following the previous point, the authors mentioned in the supplementary that InfiniteNature [23] is not compared because the training code is not available. However, to my knowledge, the testing code is already available online and comparison is possible. For example, even GeoFree [40] compares with InfiniteNature. I suggest adding InfiniteNature (or a warp+inpainting method) as a baseline if there is no strong reason why this cannot be done.

3. If explicitly rendered feature and implicitly rendered feature are complementary, then why not just manually fuse the two features according to the out-of-view mask or send the mask as an additional input to the final decoder so that the decoder can learn to fuse them?  In this case, the two features do not have to mimic each other. Thus, I doubt if the transformation similarity loss is necessary. I suggest adding this experiment as an ablation study.

Some typos and grammar issues:
Line 53: a novel loss function that explicit features improve ... -> a novel loss function that motivates explicit features to improve ...
Line 116: should $I_{ref} \in \mathcal{R}^{H\times W \times C}$ be $I_{ref} \in \mathcal{R}^{H\times W \times 3}$?
Line 135: resulting in a space complexity is reduced from ... -> resulting in a space complexity reduction from ...
Line 207: Therefore, We -> Therefore, we


Considering all the strengths and weaknesses, I rate borderline reject now. I may change my rating if my concerns in weaknesses are addressed.

=============

I have read the authors' feedback. My major concerns on InfiniteNature and feature fusion are well addressed. Thus, I increase my rating to weak accept.

---

> ### Author Response · Authors · 2022-08-02
> **Response to Reviewer qWX6 (1)**
>
> Dear Reviewer qWX6,
>
> We deeply appreciate the valuable comments by Reviewer qWX6. The valuable comments are helpful in improving our manuscript. We will address all concerns raised by the reviewer and revise the paper accordingly.
> ***
> ### Warping and synthesizing the out-of-view pixels is a straightforward way for the seesaw problem.
> Explicit methods leverage 3D inductive biases to guide the view synthesis network to preserve 3D transformed contents, and various generative models are applied to complete the unseen regions. In small view changes, explicit methods can produce high-quality images because the content of the reference viewpoint still occupies a large portion. However, filling unseen regions with semantically coherent pixels in large view changes remains a challenge for explicit methods, where unseen regions occupy a large portion. For one reason, the 3D transformed image loses some parts of the reference image, so painting from the 3D transformed image may not fully leverage information from the reference image.
>
> Meanwhile, implicit methods synthesize the target view image given the reference image and camera parameters. Less enforcing 3D inductive bias and using powerful autoregressive transformers, implicit methods can synthesize diverse and realistic out-of-view regions but fail to preserve seen regions.
>
> The approach of warping and synthesizing the out-of-view pixels like inpainting has been explored a lot in explicit methods.
> - SynSin uses a spatial feature predictor to extract feature maps and reproject feature maps to the target view by a differentiable renderer and a depth regressor. Then, given reprojected feature maps, a refinement network should inpaint the out-of-view and generate synthesized images.
> - PixelSynth uses a depth module and a projector to reproject a reference image and fill the out-of-view regions with an autoregressive outpainter. Then, a refinement module is used to improve image quality. With an autoregressive outpainter, PixelSynth improves the generation ability of unseen regions but is still worse than the implicit method.
>
> Likewise, InfiniteNature adopts the approach of reprojecting and generating the out-of-view regions and belongs to the explicit method. As mentioned above, explicit methods lack the ability to generate out-view-regions, and it was not fully designed for solving the seesaw problem.
>
> Also, one can say that a better generation model can improve this ability. Because previous explicit methods have failed this, we partially agree on this point of view. However, obtaining generation ability for unseen regions from implicit method is a straightforward and effective approach for complementing explicit methods. From this, our method is much more suitable for solving seesaw problems than InfiniteNature.
>
> ***
> ### Add warp + inpainting methods and InfiniteNature as baselines.
> As mentioned above, SynSin and PixelSynth adopt the approach of warping and inpainting and are used as the baseline of our method. We also report the results of InfiniteNature in Table 2 in the main paper. The results show that our method consistently outperforms InfiniteNature by comparing FID and PSNR. Since we adopt the train and test split of GeoFree which is different from InfiniteNature, and their training code is not available, we can not fairly compare InfiniteNature with baselines and our methods. We clearly stated this point with the results of InfinteNature in the revised manuscript.

---

> > ### Author Response · Authors · 2022-08-02
> > **Response to Reviewer qWX6 (2)**
> >
> > ### Ablation study in which the final decoder fuses two features using the out-of-view mask.
> > Thank you for your valuable suggestions. We use the out-of-view mask $\textbf{O}(p)$ as an additional input for the encoder by concatenating explicitly and implicitly rendered features without transformation similarity loss $L_{ts}$. We conducted experiments on RealEstate10K, and Table below shows the results. The results show that additional mask information achieves slight improvements for PSNR and FID compared to the model without $L_{ts}$. Nevertheless, the additional out-of-view mask does not improve performance as dramatically as transformation similarity loss. This implies that our transformation similarity loss is strongly effective in complementing explicit and implicit branches. Through this ablation study suggested by Reviewer qWX6, we can highlight the importance of transformation similarity loss. We add details and results in Appendix C.1 in the revised supplementary material.
> > |            Model           |   Small   |           |   Medium  |           |   Large   |           |  Average  |           |
> > |:--------------------------:|:---------:|:---------:|:---------:|:---------:|:---------:|:---------:|:---------:|:---------:|
> > |                            |  PSNR-vis |    FID    |  PSNR-vis |    FID    |  PSNR-vis |    FID    |  PSNR-vis |    FID    |
> > |         No $L_{ts}$        |   16.55   |   35.52   |   15.86   |   38.10   |   15.30   |   47.74   |   15.90   |   40.45   |
> > | $\textbf{O}(p)$ as feature |   16.86   |   34.74   |   15.92   |   36.10   | **15.36** |   46.43   |   16.05   |   39.09   |
> > |            ours            | **16.94** | **32.42** | **15.97** | **33.04** | **15.36** | **35.26** | **16.09** | **33.57** |
> > ***
> > ### Add some video results.
> > We sincerely appreciate Reviewer qWX6’s recommendation. Videos moving in the 3D world from a single image are so interesting and intuitive for single-image view synthesis. When we build our project page on github, we will include the video results.
> > ***
> > ### Some mismatch in reprojected regions.
> > As we trained the depth estimation network in a self-supervised manner, some reprojected regions may not be accurate. During training, the model learns to fix inaccurate reprojected regions balancing with the implicit transformation branch. Therefore, the model does not completely preserve the reprojected regions and causes some mismatch. Nevertheless, our method consistently outperforms baselines in PSNR measure only in the reprojected area with trained Monodepth2 (see Appendix C.1 and Table 1-2 in the revised supplementary material) and shows better qualitative results (see Appendix C.2), showing better preserving ability of seen pixels than previous methods.
> > ***
> > ### Typos
> > We thank the reviewer for pointing out typos. We corrected it.

---

> > > ### Comment · Reviewer_qWX6 · 2022-08-03
> > > **Response to the authors**
> > >
> > > Thank you for your detailed feedback. My major concerns on InfiniteNature and feature fusion are well addressed. Thus, I increase my rating to weak accept.

---

> > > > ### Author Response · Authors · 2022-08-04
> > > > **Response to Reviewer qWX6**
> > > >
> > > > We are glad that your concerns have been resolved. Thank you for taking the time to respond to the rebuttal.

---

### Official Review · Reviewer_mgev · 2022-07-10

**Rating:** 6
**Confidence:** 4
**Soundness:** 2 fair
**Presentation:** 3 good
**Contribution:** 3 good

**Summary:**

There are two main paradigms in the task of single-image view synthesis. The explicit methods use the explicit 3D geometry and thus achieve good results on the reprojected pixels, while the implicit methods leverage implicit 3D inductive bias to implicitly learn the 3D geometry and could complete realistic out-of-view image regions. However, none of them can reach both goals. To mitigate such a seesaw problem, this paper bridges both implicit and explicit geometric transformations in a unified framework. Besides, a novel loss is proposed to fuse the geometry features from implicit and explicit branches. The proposed method can render state-of-the-art results on the novel views and achieve about 100x speedup in rendering compared to implicit autoregressive models.

**Questions:**

In this manuscript, experiments are not comprehensive. Please refer to the questions proposed in the weaknesses.

**Limitations:**

1. The idea is a little bit straightforward, just like a simple combination of A and B.
2. Experiments are not comprehensive and not enough to demonstrate the effectiveness of the proposed method.
3. Some confusing sentences inside:
- Lines 50 ~ 51: Our approach consists of architecture and loss functions.
- Lines 159 ~ 160: ... generating photo-realistic images. What's the relationship between rendering speed and image quality. The emphasis on "photo-realistic" is unnecessary.
- Line 48: application area -> application areas
...

**Strengths And Weaknesses:**

Pros:
1. Combining the explicit and implicit geometry features in a unified framework is a reasonable and straightforward idea. Since explicit 3D geometry can preserve reprojected image areas and implicit 3D geometry helps to complete out-of-view regions. Meanwhile, the experiments on RealEstate10K and ACID datasets prove the effectiveness of such a combination.
2. The whole paper is succinct and easy to follow.
3. The ablation study in Sec. 4.3 & 4.4 demonstrates the necessity of the proposed Transformation Similarity Loss.

Cons:
1. From the main quantitative results in Table 2, the proposed method only shows superiority in terms of the FID metric. Although using both explicit and implicit geometric transformations, this method still works worse than SynSin. It seems that this method doesn't solve the aforementioned trade-off. For the lower FID score, this method just equips with a better imagination ability, which may be boosted with a more powerful adversarial loss. In lines 164~166, the authors mention they use a new GAN loss. How about the performance of SynSin + this new loss? Such a very important ablation study is not included in this manuscript.
2. The paper only provides the results on RealEstate10k and ACID datasets. How about the results on Matterport3D?
3. Given the better performance on the FID score, this method should compare with another two crucial baselines, Infinite Nature and Look Outside Room. I notice the authors' clarification in the supp. material. If available, the performance comparisons must be provided.

---

> ### Author Response · Authors · 2022-08-02
> **Response to Reviewer mgev**
>
> Dear Reviewer mgev,
>
> We sincerely appreciate the positive and insightful review by Reviewer mgev, _e.g., combining the explicit and implicit geometry features in a unified framework is a reasonable and straightforward idea, easy to follow, and the ablation study demonstrates the necessity of the proposed Transformation Similarity Loss_. They were very helpful in refining our manuscript. We will address all raised concerns by the reviewer and revise the paper accordingly.
> ***
> ### Method only shows superiority in terms of the FID metric.
> Thank you for pointing this out. As shown in Table 2, our method achieves slightly worse PSNR than SynSin in the large splits. This is because PSNR is measured on the entire image which includes the unseen regions. To more accurately compare performance to preserve seen pixels, we report PSNR on visible pixels using the retrained Monodepth2 on RealEstate10K and ACID, respectively. The PSNR-vis column in Table below shows the results. In these results, our method outperforms SynSin in all splits, showing better preserving seen regions. The results and details are shown in Appendix C.1 in the revised supplementary material.
> We agree that a new GAN loss can lower the FID score than SynSin, and investigating the impact of the new GAN loss is an important ablation study. We report the performance of SynSin trained with this new loss on the RealEstate10K dataset in Table below.
> |    | Small     |  | Medium    | | Large     |           |
> |-|-|-|-|-|-|-|
> | Method           | PSNR-vis  | FID       | PSNR-vis  | FID       | PSNR-vis  | FID       |
> | SynSin                 | 15.67     | 41.75     | 15.46     | 43.06     | 15.28     | 61.67     |
> | SynSin + our $L_{adv}$ | 15.45     | 40.43     | 15.31     | 39.13     | 15.09     | 54.27     |
> | Ours (w. $L_{ts}$)     | **16.94** | **32.42** | **15.97** | **33.04** | **16.09** | **33.57** |
>
>
> Combining SynSin and GAN loss used in our method lowers the FID score in all splits. This shows that a new GAN loss improves the performance of the generation. However, it is still a worse FID score than our method. From this, we want to emphasize that our method is not just boosted with a more powerful adversarial loss. Our architecture advances bridging explicit and implicit geometric transformations with transformation similarity loss contributes significantly to performance gain. Also, the new GAN loss does not solve the seesaw problem as it improves SynSin in FID by sacrificing PSNR-vis. This ablation study helps us to highlight the effectiveness of bridging both transformation branches. We sincerely appreciate and add these experiments in Appendix C.1 of the revised supplementary material.
> ***
> ### Compare with InfiniteNature and LookOutsideRoom.
> As mentioned in our manuscript, the code of LookOutsideRoom was not available at the initial submission, but it is available now. We add the results of LookOutsideRoom to the revised supplementary material. The evaluation code of InfiniteNature is available, but training codes are not. Since we adopt the train and test split of GeoFree which is different from InfiniteNature, we can not fairly compare InfiniteNature with baselines and our methods. We state this point and attach the results to Table 2 in the main paper. The results show that our method outperforms InfiniteNature and LookOutside room consistently in all splits.
> ***
> ### Provide the results on Matterport3D for comprehensive experiments.
> We appreciate your suggestion. We agree that results on Matterport3D can improve the manuscript. However, In the rebuttal phase, we requested to download Matterport3D, but we have not received a response. If we can access the dataset, we will add the results to the manuscript in matterport3D.
> Even without the results of Matterport3D, we demonstrate the superiority of our method with representative indoor and outdoor datasets. If Reviewer mgev feels that our experiments are not comprehensive due to a lack of results in Matterport3D, please consider that GeoFree conducted experiments on the same dataset as ours. Furthermore, GeoFree, LookOutsideRoom and PixelSynth validate their effectiveness on two datasets as same as ours.
> ***
> ### The idea is a little bit straightforward, just like a simple combination of A and B.
> The major point of our methods is bridging explicit and implicit transformation branches. We want to emphasize that our method is not a simple combination of explicit and implicit transformation branches. We go beyond the simple connection of two branches by proposing transformation similarity loss which complements each other. As the reviewer agrees, transformation similarity loss is necessary for connecting two branches. We would be very grateful if reviewer mgev would consider that our method is a combination of implicit, explicit branches and transformation similarity loss.
> ***
> ### Some confusing sentences.
> We thank the reviewer for pointing out some confusing sentences. We corrected it.

---

> > ### Comment · Reviewer_mgev · 2022-08-07
> > **Response to the authors**
> >
> > Thanks for the authors' comments and clarifications. This paper turns much more complete with these added experiments, which are suggested to be contained in the revision. Most of my concerns are resolved and I decide to raise my rating accordingly.

---

> > > ### Author Response · Authors · 2022-08-08
> > > **Response to Reviewer mgev**
> > >
> > > We are glad that Reviewer mgev's concerns have been resolved. Thank you for taking the time to respond to the rebuttal.

---

### Official Review · Reviewer_k3Jg · 2022-07-10

**Rating:** 7
**Confidence:** 4
**Soundness:** 3 good
**Presentation:** 4 excellent
**Contribution:** 3 good

**Summary:**

This paper studies the task of single image novel view synthesis. The key message is very clear. Methods with explicit 3D projection mechanism, like SynSin, better preserve observed regions, whereas methods with implicit renderer, like GeoFree, do better at outpainting (synthesizing unseen regions). And using both of them in the same framework can complement and reinforce each other and get the best of both worlds.

The paper does a very good job at conveying this message, and presents promising results on both indoor and outdoor datasets to demonstrate the benefit of this joint approach.

**Questions:**

1. Report PSNR both only on seen pixels and on the entire image, if it's manageable during the rebuttal.
2. Finetune the models of the previous methods on the same dataset, if this is not done currently.
3. Show more diverse visual comparisons, maybe in the final version.

**Limitations:**

The paper includes a brief discussion on the limitations. The current quality of the results are still quite limited. It would be helpful to provide some failure examples if any.

**Strengths And Weaknesses:**

## Strengths
### S1 - Message is clear and makes a lot sense
- The paper delivers the motivation of using both implicit and explicit rendering in single image NVS in a very clear and concise manner.
- The benefits of both worlds can be maintained by defining losses that use the better one to guide the other, ie, features from the explicit branch will guide the ones from implicit branch in observed regions, with the gradient of the former being detached, and the other way around in unobserved regions.
- The resulting model seems to overcome the shortcomings of both approaches, as shown in the comparison results.

### S2 - Clear and concise writing
- I enjoy reading the paper. It's very clear and coherent.

## Weaknesses
### W1 - Results are still a bit unsatisfactory
- First, I was expecting more qualitative comparisons in the supplementary material. There are only four examples shown. They're all good examples with large viewpoint changes. But showing more diverse results will be helpful.
- Based on the limited comparisons provided, it does not seem that the proposed method can fully preserve the benefits of both implicit and explicit methods. The quality of both observed and unobserved (outpainted) regions seems to be slightly compromised.
  1. Observed regions do not seem to be preserved very well, seemingly worse than SynSin and PixelSynth on indoor scenes. I think neither of them has been trained on images of outdoor scenery, so I'm not surprised they both fail on scenery images. The PSNR is also worse than SynSin with larger viewpoint changes.
  2. Outpainted regions also appear worse than GeoFree, although the latter uses an autoregressive framework.
  3. There seem to be obvious artifacts in the synthesized images, eg the center of the generated image in second row of Fig 4 (dinning table).
- Are the PSNR computed only on the seen pixels or on the entire image? Reporting both numbers will resolve the question in 1.
- Have the compared models been finetuned on the same dataset? If not, I do not think the comparison is fair.

### Minor comments
- Line 116: I suppose $f_0$ has a larger channel dimension than input image $I_{ref}$. Use different symbols for their channel dimensions.

---

> ### Author Response · Authors · 2022-08-02
> **Response to Reviewer k3jg**
>
> Dear Reviewer k3jg,
>
> We truly appreciate the positive and thoughtful review by Reviewer k3jg, _e.g., the message is clear and makes a lot of sense and concise writing_. We found them very helpful in improving our manuscript. We will address all raised concerns by the reviewer and revise the paper accordingly.
> ***
> ### Report PSNR both on seen pixels and on the entire image.
> Since RealEstate10K and ACID datasets do not have ground-truth depth, we can not accurately calculate the observed and unobserved regions. Therefore, we did not report PSNR on observed pixels in the initial submission. We agree that this induces the reviewer's question of whether observed regions do not seem to be preserved well.
>
> We report PSNR on seen pixels that are obtained from trained Monodepth2 on RealEstate10k and ACID, respectively. Table below shows the results. Our method consistently outperforms baselines by comparing PSNR on seen pixels, showing better preserving contents on observed regions. These results differ from the results in Table 2 that our method is slightly worse than SynSin in large splits when PSNR is measured on the entire image. By measuring PSNR on seen pixels, we can highlight that our method well preserves seen pixels. We appreciate the reviewer's recommendation and add the details and results to Appendix C.1 in the revised supplementary material.
> |       RealEstate10K          |||| |               ACID          |||||
> |-|-|-|-|-|-|-|-|-|-|
> | Methods         |   Small   |   Medium  |   Large   | Average | Methods                 |   Small   |   Medium  |   Large   |  Average  |
> | Tartarchenko    |   11.16   |   10.75   |    10.70   |  10.87  | Tatarchenko             |   14.53   |   14.34   |   14.62   |    14.50   |
> | Viewappearance  |   12.39   |   12.89   |    12.50   |  12.59  | Viewappearance          |   14.66   |   13.76   |   13.22   |   13.88   |
> | SynSin|   15.67   |   15.46   |   14.72   |  15.28  | SynSin                  |   18.05   |   17.16   |   17.32   |   17.51   |
> | SynSin-6x|   15.43   |   15.54   |   14.92   |   15.30  | InfiniteNature (1-step) |   16.97   |   15.74   |   15.24   |   15.98   |
> | PixelSynth|   15.62   |    15.60   |   14.64   |  15.29  | InfiniteNature (5-step) |   15.76   |   15.44   |   15.62   |   15.61   |
> | GeoFree|   14.89   |   14.37   |    13.60   |  14.29  | PixelSynth              |   17.61   |   16.22   |   15.32   |   16.38   |
> | LookOutSideRoom |     12.78    |     13.13     |     12.54     |    12.82    | GeoFree                 |   15.26   |   14.86   |   14.67   |   14.93   |
> | Ours| **16.94** | **15.97** | **15.36** |  **16.09**  | Ours| **18.17** | **17.58** | **17.88** | **17.88** |
> ***
> ### Finetune the models of the previous methods on the same dataset.
> In all our experiments, RealEstate10K and ACID datasets are used. Because these datasets share data with YouTube links and YouTube pages can be deleted, some video sequences can be missed in our experimental settings. For the models of previous methods where pretrained models are available, we retrained these models in our experimental settings and observed that retrained models perform slightly worse or similar to pretrained models. Therefore, we used pretrained models if they were available. In cases where pretrained models are not available such as SynSin and PixelSynth trained on ACID, etc., we retrained them and used them as the baseline. In this way, we conducted a fair comparison.
> ***
> ### Show more diverse visual comparisons.
> We agree with the suggestion that more diverse qualitative comparisons are helpful in improving our manuscript. We report more qualitative comparisons according to the out-of-view ratio on RealEstate10K and ACID. Each example includes a warped image, a target image and PSNR-vis scores, making it easy to see how well the model preserves the seen contents and completes the unseen contents. For diversity, we report various cases, such as _going straight_ or _outdoor synthesis_ in RealEstate10K and an example with black above and below the image in ACID. The results are shown in Appendix C.2 in the revised supplementary. The results show that our method well preserves observed regions and fills unseen regions than baselines.
> ***
> ### Provide some failure examples and artifacts.
> Following the reviewer’s suggestion, we investigated the failure examples. Since the depth estimation network is trained in a self-supervised manner, there are some errors on depth maps, resulting in reprojected regions being mismatched with the target image. We observed that artifacts, as seen in the second row of figure 4, occur in incorrectly reprojected areas. We conjecture that artifacts appear when the correcting procedure fails internally because the model learns to fix erroneously reprojected regions by leveraging implicit branches during training. We add the discussion in Appendix D of the revised supplementary material.
> ***
> ###  Minor comments
> We fix the channel dimension of $I_{ref}$ to three.

---

> > ### Comment · Reviewer_k3Jg · 2022-08-09
> > **Response**
> >
> > I appreciate the authors' detailed response. I like the new evaluation on the seen pixels, and the additional qualitative results in the supp mat. The results look pretty convincing. I have no further questions and will keep my recommendation to accept the submission.

---

> > > ### Author Response · Authors · 2022-08-09
> > > **Response to Reviewer k3jg**
> > >
> > > We really appreciate the valuable and insightful comments of Reviewer k3jg that strengthen our manuscript. Thank you for taking the time to respond to the rebuttal.

---

### Author Response · Authors · 2022-08-02
**Initial Revision.**

We sincerely appreciate constructive reviews. We share details of our initial revision.

* **[Suggested by Reviewer gwT5]:** The ablation study on the type of set attention is moved to the main paper. Because of limited space, we move the limitation and potential social negative impact to Appendix D of the supplementary material.
* **[Suggested by Reviewer qWX6, mgev]**: InfiniteNature and LookOutsideRoom are modified from potential baselines to baselines:
	* We add InfiniteNature and LookOutsideRoom in Table 1 of the main paper.
	* We add the results of LookOutside on RealEstate10k and the results of InfiniteNature on ACID in Table 2 of the main paper.
	* We report the average inference time of two methods in Table 3 of the main paper.
	* We add the summary of two methods to Appendix B.2 of the supplementary material.
* **[Suggested by Reviewer k3jg, gwT5]**: PSNR measured on the seen pixels with trained Monodepth2 model is reported as PSNR-vis. We add results to Table 1, 2 in the supplementary material and update all tables in the supplementary material.
* **[Suggested by Reviewer k3jg, gwT5]**: Diverse qualitative results are added to Appendix C.2 of the supplementary material.
* **[Suggested by Reviewer mgev]**: The results of SynSin + our GAN loss on RealEstate10k are added to Appendix C.1 of the supplementary material.
* **[Suggested by Reviewer qWX6]**: We add an ablation study on the mask as an additional input to the final decoder to Appendix C.1 of the supplementary material.
* We corrected typos.

---

### Author Response · Authors · 2022-08-09
**Thanks for the discussion.**

Dear reviewers,

We deeply appreciate your time and effort to review our manuscript.

As reviewers highlighted, we believe that our work proposes a novel and well-motivated method to address the seesaw problem of existing methods, i.e a trade-off between preserving observed regions and synthesizing unseen regions in the target viewpoint, with clear presentation.

To best respond to the questions and concerns raised by the reviewers, we have carefully revised and enhanced our manuscript with additional experiments and more detailed results.

The main question and concerns from the reviewers are addressed as follows:
* **[Reviewer qWX6, mgev] - More Baselines:** Reviewer qWX6, mgev suggested adding baselines. We added InfiniteNature and LookOutsideRoom as baselines and observed that our method still outperforms their methods.
* **[Reviewer k3jg, gwT5] - PSNR on reprojected pixels:** As suggested, we verify the quality of preserving reprojected regions by evaluating PSNR measured on the reprojected pixels with the trained Monodepth2 model, which is denoted as PSNR-vis. Our method outperforms previous methods on PSNR-vis regardless of view changes.
* **[Reviewer k3jg, gwT5] - Diverse Qualitative Results:** As described in Appendix C.2, we report more qualitative comparisons according to the out-of-view ratio, and our method significantly better preserves reprojected regions and completes out-of-view areas. Besides, Reviewer k3jg comments additional qualitative results look pretty convincing, and Reviewer gwT5 also comments the new evaluation results look good.
* **[Reviewer mgev, qWX6] - More Ablation Studies:** As described in Table 4-5 of the supplementary material, we further conducted two ablation studies to verify our method. The results of SynSin with our GAN loss demonstrate that explicit methods can be improved by using better generative models, but they also have the seesaw problem. This ablation study highlights the effectiveness of bridging both transformation branches. Additionally, the out-of-view mask does not dramatically improve performance compared to our transformation similarity loss when it is used as additional input to the final decoder. This shows that our transformation similarity loss complements explicit and implicit branches quite substantially.
* **Clarification:** We clarified concerns for fair comparison (Reviewer k3jg), seesaw problem (Reviewer qWX6), simplicity of our approach (Reviewer mgev), and definition of the explicit methods (Reviewer gwT5).

We sincerely appreciate that most of the reviewers’ concerns were well addressed and all of the reviewers have been positive about our submission. With valuable feedback from the reviewers, we sincerely believe that the revised manuscript could deliver the benefits of our method to the audience of NeurIPS.

Best regards,

Authors.

---

### Meta-Review · Area_Chair_MXFt · 2022-08-26

**Recommendation:** Reject
**Confidence:** Certain

**Metareview:**

The reviewers found the approach of combining implicit and explicit methods for single-view synthesis both interesting and novel. All four reviewers recommended accepting the paper. You addressed many of the remaining questions/concerns in your rebuttal and in the author-reviewer discussion. Please, go through the reviews once more and make sure you have included the changes in the final version of the paper.

Update: given the extent of revisions required, we agreed that this would require another full evaluation of the paper due to the substantial changes that would need to be made to this paper. We hope that the rebuttal would help to guide subsequent submissions of this manuscript

**Award:**

No

---

### Decision · Program_Chairs · 2022-09-14

Reject